# Amyloid-β42/40 ratio drives tau pathology in 3D human neural cell culture models of Alzheimer's disease

Sang Su Kwak [1,9], Kevin J. Washicosky[1,9], Emma Brand[1], Djuna von Maydell[1], Jenna Aronson[1,2], Susan Kim[1], Diane E. Capen[3], Murat Cetinbas[4], Ruslan Sadreyev[4], Shen Ning[1,5], Enjana Bylykbashi[1], Weiming Xia[6,7], Steven L. Wagner[8], Se Hoon Choi[1], Rudolph E. Tanzi [1✉] & Doo Yeon Kim [1✉]

The relationship between amyloid-β (Aβ) species and tau pathology in Alzheimer's disease (AD) is not fully understood. Here, we provide direct evidence that Aβ42/40 ratio, not total Aβ level, plays a critical role in inducing neurofibrillary tangles (NTFs) in human neurons. Using 3D-differentiated clonal human neural progenitor cells (hNPCs) expressing varying levels of amyloid β precursor protein (APP) and presenilin 1 (PS1) with AD mutations, we show that pathogenic tau accumulation and aggregation are tightly correlated with Aβ42/40 ratio. Roles of Aβ42/40 ratio on tau pathology are also confirmed with APP transmembrane domain (TMD) mutant hNPCs, which display differential Aβ42/40 ratios without mutant PS1. Moreover, naïve hNPCs co-cultured with APP TMD I45F (high Aβ42/40) cells, not with I47F cells (low Aβ42/40), develop robust tau pathology in a 3D non-cell autonomous cell culture system. These results emphasize the importance of reducing the Aβ42/40 ratio in AD therapy.

[1] Genetics and Aging Research Unit, MassGeneral Institute for Neurodegenerative Disease, Department of Neurology, McCance Center for Brain Health, Massachusetts General Hospital, Harvard Medical School, Charlestown, MA 02129, USA. [2] Department of Brain and Cognitive Sciences, Massachusetts Institute of Technology, Cambridge, MA 02139, USA. [3] Center for Systems Biology and Program in Membrane Biology, Division of Nephrology, Massachusetts General Hospital, Harvard Medical School, Boston, MA 02114, USA. [4] Department of Molecular Biology, Massachusetts General Hospital, Boston, MA 02114, USA. [5] Graduate Program for Neuroscience, Boston University School of Medicine, Boston, MA 02118, USA. [6] Geriatric Research Education and Clinical Center, Edith Nourse Rogers Memorial Veterans Hospital, Bedford, MA 01730, USA. [7] Department of Pharmacology and Experimental Therapeutics, Boston University School of Medicine, Boston, MA 02118, USA. [8] Department of Neurosciences, University of California, San Diego, La Jolla, CA 92093, USA. [9] These authors contributed equally: Sang Su Kwak, Kevin J. Washicosky. ✉email: tanzi@helix.mgh.harvard.edu; dkim@helix.mgh.harvard.edu

Aβ42 is a key mediator of Alzheimer's disease (AD) pathogenesis based on evidence from genetic, biochemical, and cell biological studies[1]. The vast majority of known familial AD mutations (>200) increase the Aβ42/40 ratio, which strongly supports that Aβ42 and its relative ratio to Aβ40 play an important role in AD pathogenesis (http://www.alzforum.org/mutations)[2,3]. Recent comprehensive studies with FAD patient-derived human neuronal cells also confirmed the consistent increases in Aβ42/40 ratio in human neurons harboring different FAD mutations, re-emphasizing the important pathogenic role of Aβ42/40 ratio in AD[4]. However, current AD patient-derived human neural cells and transgenic mouse models have failed to show that Aβ42 and/or the high Aβ42/40 ratio directly cause tau pathology, including NFTs, a key pathological hallmark of AD.

Transgenic AD mice carrying single or multiple FAD mutations in the amyloid-β precursor protein (*APP*) and/or presenilin 1 (*PSEN1*) genes have been used as a standard AD model to study Aβ plaques and Aβ-driven synaptic/memory deficits. However, these mice do not fully develop NFTs or robust neurodegeneration[5–7]. BRI-Aβ42-40 transgenic mouse models have provided valuable initial insight into the critical role of Aβ42, Aβ40, and Aβ42/40 ratio on regulating Aβ aggregation in brains[8]. However, BRI-Aβ42 and BRI-Aβ42-Aβ40 transgenic mouse models do not clearly demonstrate Aβ42-driven cognitive deficits or tau pathology[8,9]. Only triple transgenic mouse models show both β-amyloid plaques and NFTs. However, NFT pathology in these models depends mainly on the overexpression of frontotemporal lobar degeneration (FTLD)-associated human tau protein with mutations in the *MAPT* gene[10–12], which have not been associated with AD. Thus, current mouse models cannot provide comprehensive information regarding Aβ42-driven pathogenic cascades leading to NFTs and neurodegeneration.

AD patient-derived human neurons have been used as an alternative model system to test the impact of Aβ42 on NFT pathology with endogenous human tau proteins. However, the tau pathology observed in these AD neurons has not been shown to be regulated by either Aβ42 or the Aβ42/40 ratio[13–16]. Additionally, the elevated total tau and p-tau in these AD neurons did not display filamentous aggregation, which is a critical marker of NFT pathology. Treatments with synthetic Aβ42 induced various neuronal deficits in human neurons, including synaptotoxicity, ER stress, and neuronal death[17–20]. However, no clear tau pathology was detected in these models and the use of synthetic Aβ42 preparation with different aggregation protocols limits interpretation of these studies together. To date, no human neuronal cell model has been developed to dissect the positive or negative roles of different Aβ species on AD pathogenesis.

Recently, we developed a 3D AD cellular model displaying both robust extracellular Aβ deposits (Aβ plaques) and Aβ-driven tau pathology, including somato-dendritic accumulation of p-tau and detergent-insoluble/silver-stained intracellular tau aggregation leading to neurofibrillary tangles (NFTs) and paired-helical filaments (PHFs)[21,22]. In this model, overexpression of human *APP*^Swedish/London (APPSL) and *PSEN1*^ΔE9 (PS1ΔE9) robustly increased the levels of pathogenic Aβ peptides, including Aβ42 in 3D-differentiated ReNcell VM cells, immortalized hNPCs derived from human fetus[21–24]. The expression levels of APPSL determine total Aβ levels while PS1ΔE9 levels affect Aβ42/40 ratio by altering PS1/γ-secretase complex in AD hNPCs (Supplementary Table 1). However, the heterogeneous nature of these FAD hNPCs, a mixture of cells expressing variable levels of APPSL and/or PS1ΔE9, makes it difficult to directly address the impact of Aβ42 levels and/or Aβ42/40 ratio on Aβ-driven tau pathology and possibly neurodegeneration.

Here, we comprehensively assessed the impact of Aβ42/40 ratio on AD pathogenesis using advanced 3D human neural cell culture models with single-cell-derived clonal hNPCs. In addition to clonal FAD hNPCs overexpressing APPSL and/or PS1ΔE9, we also developed clonal hNPCs with mutations in the APP transmembrane domain (TMD), APP TMD I45F and I47F, to compare the impact of the Aβ42/40 ratio on tau pathology in the absence of PS1ΔE9 overexpression. Using these unique cellular models, we demonstrate the critical role of the Aβ42/40 ratio on tau and NFT pathology in human neural cells. We also demonstrate that a high Aβ42/40 ratio is able to induce robust tau phosphorylation even in naïve hNPC-derived neurons lacking FAD mutations using a 3D non-cell-autonomous cell culture model.

## Results

**Clonal FAD hNPCs generated by single cell sorting**. Heterogeneous FAD hNPCs were generated as previously described by expressing lentiviral DNA constructs harboring APPSL and/or PS1ΔE9 (Supplementary Table 1)[21,22]. To obtain clonal FAD hNPCs, heterogeneous FAD hNPCs were sorted into single cells using a FACS-assisted single cell sorting protocol based on GFP and/or mCherry signals (Fig. 1a, b and Supplementary Fig. 1). Viable cells occupying individual wells in 96-well plates were expanded to form colonies within 2–3 weeks (Fig. 1c). For slow growing colonies, we added naïve ReNcell VM cells to accelerate cell proliferation. Then, hNPCs with GFP and/or mCherry expressions were enriched again by an additional FACS-assisted bulk cell enrichment protocol (Fig. 1a). Next, we analyzed secreted Aβ levels of clonal control and FAD hNPCs by Western blot analysis. As expected, clonal hNPCs produced different amounts of Aβs and sAPPαs due to differential APPSL expression in each clonal cell (Fig. 1d and Supplementary Fig. 2). Figure 1e and Supplementary Table 1 show the representative clonal control and FAD hNPCs used in this study.

As shown in Supplementary Table 1, APPSL expression is tied with GFP since they are under the same transcriptional regulation through an IRES element in ReN-mGAP cells. The same linkage exists between mCherry and PS1ΔE9. Therefore, GFP and mCherry signals in mixed and clonal ReN-mGAP AD cells can be interpreted as expression markers for APPSL and PS1ΔE9 protein expression, respectively. Figure 2a shows representative images of GFP and mCherry expression in parental ReN-mGAP cells and the clonal ReN-mGAP10#D4 cells. As expected, parental ReN-mGAP cells exhibited a heterogeneous expression pattern in GFP and mCherry while the clonal ReN-mGAP10#D4 displayed much more homogeneous expression of GFP and mCherry (Fig. 2a). These results indicate that APPSL and PS1ΔE9 expression are much more homogeneous in the clonal ReN-mGAP10#D4 cells as compared to the parental ReN-mGAP cells. Western blot analysis confirmed the expression of APPSL and PS1ΔE9 in both parental and clonal AD cells (Fig. 2b). We also monitored the expression of APP by Western blot analysis and found that APP levels were much higher in clonal FAD hNPC lines as compared to heterogeneous parental ReN-mGAP cells possibly due to the homogeneous expression of APP in higher number of cell population (Fig. 2b).

Next, we analyzed levels of the major Aβ species, Aβ38, 40, and 42 as well as their ratios (Fig. 2c, d and Supplementary Fig. 3). Clonal ReN-mGAP10#D4 and ReN-mGAP2#H10 cells showed much higher Aβ42/40 ratios as compared to the heterogeneous parental ReN-mGAP cells, owing to the homogeneous expression of APPSL and PS1ΔE9 (Supplementary Table 1 and Fig. 2a). HReN-mGAP#A4H1 also showed increases in Aβ42/40 ratio as compared to the parental HReN-mGAP cells (Fig. 2c, d).

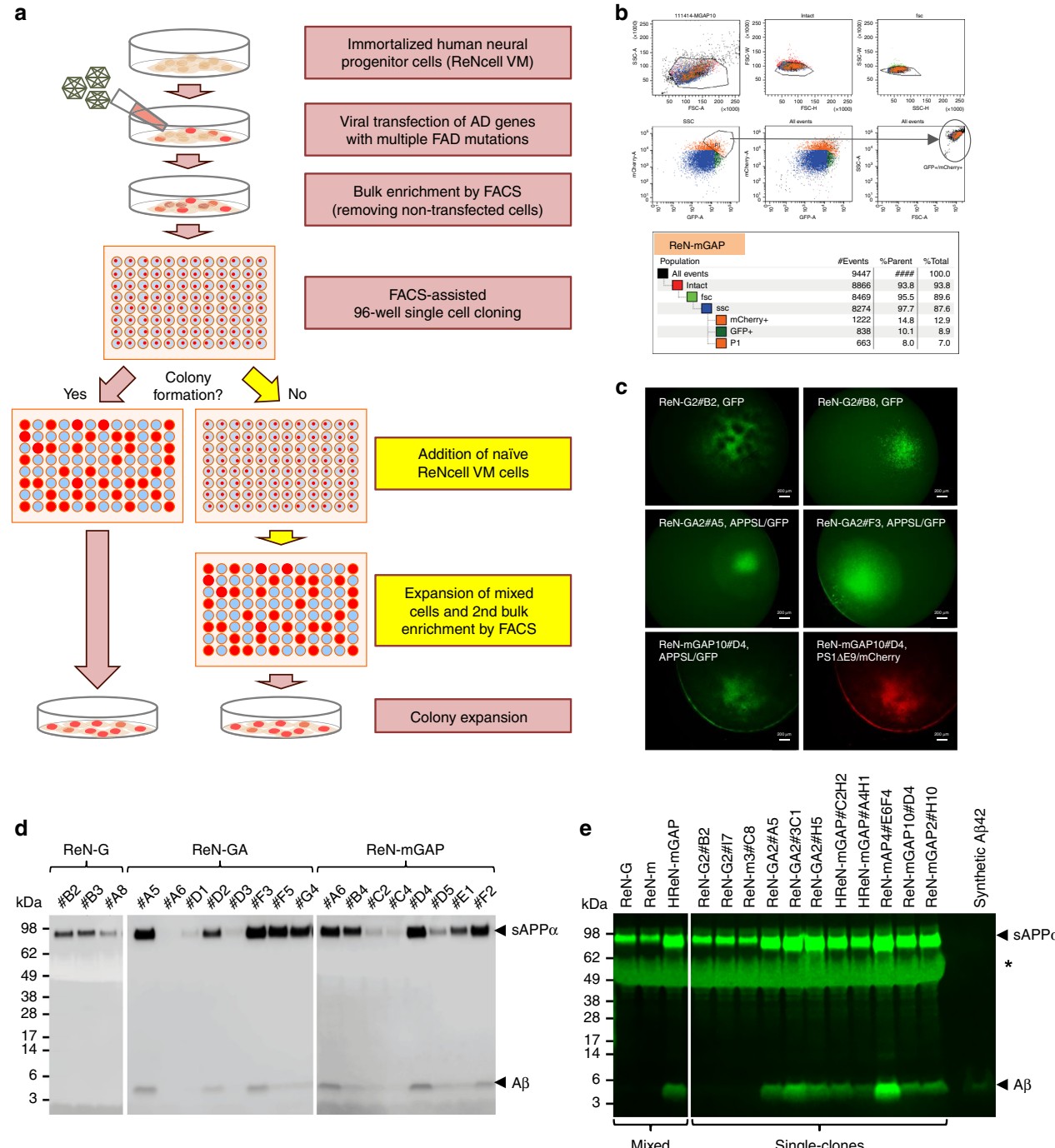

**Fig. 1 Generation of clonal control and FAD hNPCs using FACS-assisted single cell sorting. a** A workflow for preparing clonal hNPCs using FACS-assisted single cell sorting protocol. The heterogeneous FAD hNPCs were prepared by expressing lentiviral DNA constructs harboring APP$^{Swe/Lon}$ (APPSL) and/or PS1ΔE9 in naïve hNPCs (ReNcell VM). Non-AD control cells were generated by expressing lentiviral DNA constructs harboring GFP (ReN-G) or mCherry (ReN-m). In case of HReN-mGAP cells, ReN-G2 cells were infected with lentiviral particles harboring APPSL-IRES-PS1ΔE9-mCherry. The heterogeneous parental control and FAD hNPCs were subjected to FACS-assisted single cell sorting and grown for 2 weeks to develop colonies. Slow growing single clones were augmented with naïve hNPCs without expressing GFP or mCherry to facilitate cell proliferation. When the cells were confluent, fluorescence-positive clonal hNPCs were collected by an additional FACS sorting protocol. **b** Gating strategy of FACS-assisted 96-well single cell sorting for ReN-mGAP cell line. For all viable cells, mCherry fluorescence intensity (mCherry-A) was plotted as a function of GFP fluorescence intensity (GFP-A). The *P1* area was gated to select an overlapped region between high-GFP (8.9% of the GFP positive population) and high-mCherry (12.9% of the mCherry population) signals. Each individual cell within 7% of the gated population was placed into a single well of Matrigel pre-coated 96-well plates. **c** Colony formation of representative FACS-assisted clonal hNPCs in 96-well plates. Scale bars represent 200 μm. **d** Western blot analysis of Aβ levels in conditioned media from 2D-expanded clonal hNPCs derived from heterogeneous ReN-G, ReN-GA and ReN-mGAP cells. Secreted/soluble Aβs and sAPPαs were detected using anti-Aβ antibody (6E10). **e** Analysis of Aβ in media from 2D-expanded clonal FAD hNPCs. Selected clones from each parental group were grown in 6-well plates under expansion conditions. After 48 h, media was collected. Secreted/soluble Aβs and sAPPαs were detected using anti-Aβ antibody (6E10). Asterisk represents a nonspecific band.

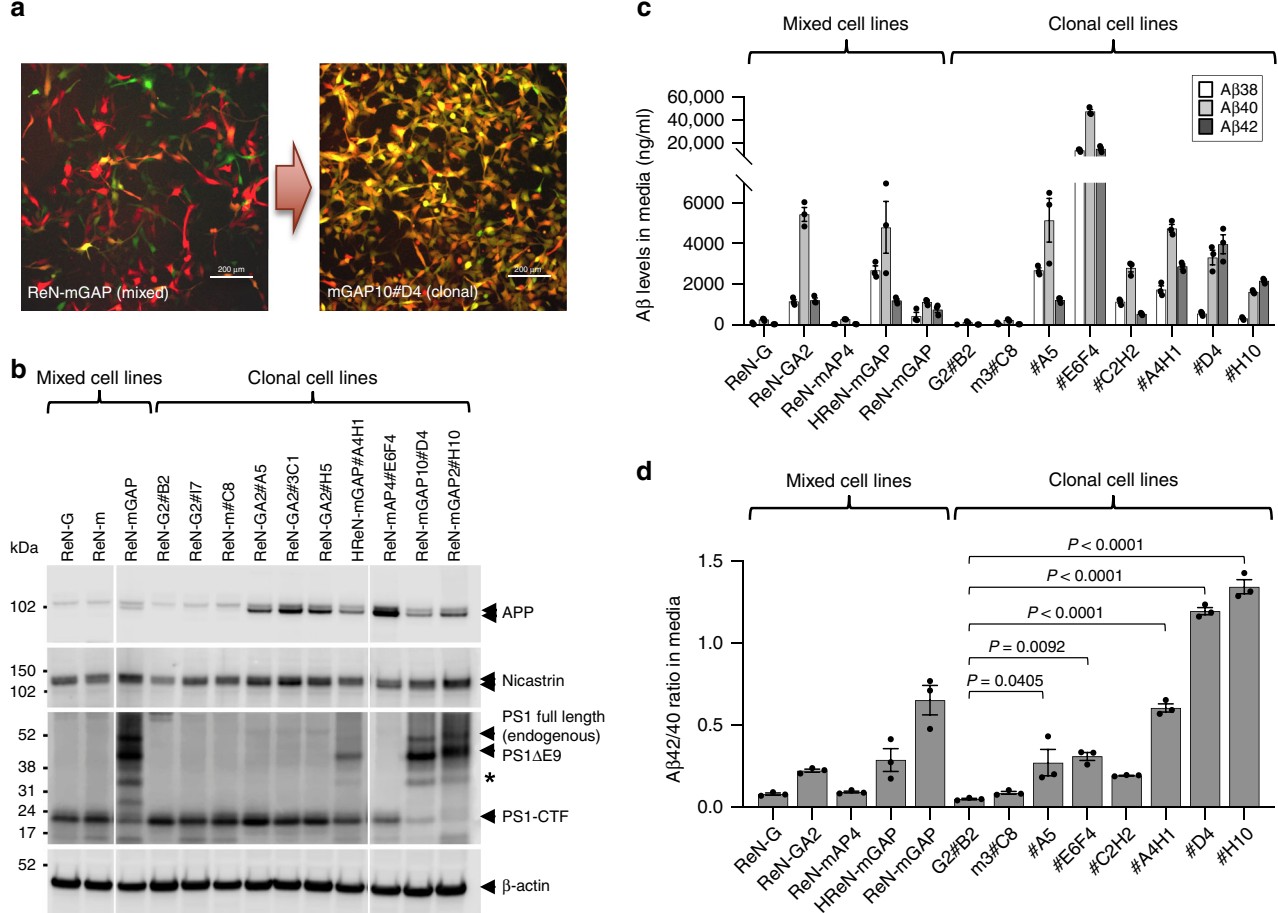

**Fig. 2 Clonal FAD hNPCs produce high levels of Aβ with different Aβ42/40 ratio. a** Representative confocal images of ReN-mGAP (mixed) and ReN-mGAP10#D4 (clonal) cells. 2D-cultured cells were expanded on glass-bottom dishes. Cells were imaged with GFP and mCherry which represent APPSL and PS1ΔE9, respectively, using confocal microscopy. Scale bars represent 200 μm. **b** Expression of APP and PS1ΔE9 in clonal hNPCs. Cells were prepared in 6-well plates and grown for 24 h. Expression levels of APP and PS1ΔE9 were monitored by Western blot analysis. Levels of nicastrin and PS1-CTF were also monitored. β-Actin was used as a loading control. Asterisk represents a nonspecific band. **c** Comparison of Aβ38, Aβ40 and Aβ42 levels between mixed and clonal hNPC lines. Indicated mixed and clonal cells (3 × 10⁶) were grown in Matrigel-coated 6-well plates for 24 h. After changing media with 1 ml of fresh expansion media, cells were incubated for an additional 24 h. Conditioned media was analyzed by MSD Aβ assay to measure concentrations of secreted Aβ species. **d** Aβ42/40 ratios were determined. All values were combined to express as mean ± SEM of three independent repeats (black dots). Statistical significances were determined by one-way ANOVA with Tukey's multiple comparisons test.

Interestingly, HReN-mGAP#C2H2, which were also derived from HReN-mGAP, showed relatively lower Aβ42/40 ratio than the parental clones while the HReN-mGAP#A4H1 from the same parental HReN-mGAP cells showed higher Aβ42/40 ratio (Fig. 2c, d). This difference between two clonal cells suggests the heterogeneous nature of parental HReN-mGAP cells. ReN-mAP4#E6F4 was the only clone from the parental ReN-mAP4 cell line that showed ~10-fold higher levels of total Aβ. Interestingly, this cell line showed much lower Aβ42/40 ratio as compared to the HReN-mGAP#A4H1 cells that express the same APPSL-PS1ΔE9 construct (Fig. 2d). Indeed, Western blot analysis showed that ReN-mAP4#E6F4 cells express very low PS1ΔE9 protein, through an unknown compensatory mechanism, which explains the reduced Aβ42/40 ratio in this cell line (Fig. 2b). Together, these data suggest that clonal FAD hNPCs can provide a stable model system to explore the robust pathogenic impact of Aβ42, Aβ40 and Aβ42/40 ratio in human neural cells.

**Clonal hNPCs differentiate into neurons and astrocytes in 3D.** Next, we differentiated the control and FAD hNPCs into neurons and glial cells under 3D culture conditions (Supplementary

Fig. 4a). 3D-differentiated control (ReN-G10) and FAD hNPCs (ReN-GA2#A5, HReN-mGAP#A4H1, ReN-mAP4#E6F4, ReN-mGAP10#D4, and ReN-mGAP2#H10) exhibited robust increases in neurite networks stained by MAP2, a marker for mature neurons, after 5 weeks (Supplementary Fig. 4b). Quantitative dot blot analysis also showed strong expression of marker proteins for neurons (Tuj1) and astrocytes (S100β and GFAP), respectively (Supplementary Fig. 4c–e). Additionally, we used mRNA-sequencing analysis to further characterize differentiation of clonal and mixed hNPCs in 3D. We determined gene expression profiles for undifferentiated naïve ReNcell VM and for differentiated mixed parental ReN-GA2, clonal FAD ReN-GA2#A5, and clonal non-FAD ReN-G2#B2 hNPCs in 3D gels for 7 weeks.

Pearson correlation and differential gene expression analysis showed a strong correlation between 3D-differentiated mixed parental (ReN-GA2) and clonal FAD hNPCs (ReN-GA2#A5) (Supplementary Fig. 5a, b). Importantly, we found robust increases in various pan and mature neuronal and glial markers following differentiation in 3D, including MAP2 (Pan-neuronal), GABBR2 (GABAergic), NMDAR2B (glutamatergic), GFAP (astrocytic) and PLP1 (oligodendrocytic) (Supplementary Fig. 5c).

These data clearly demonstrate that both clonal control and FAD hNPCs undergo similar neural differentiation as we previously showed using mixed control and FAD hNPCs[21,22].

**Aβ42/40 ratio determines Aβ aggregation.** Previously, we showed that robust accumulation/aggregation of Aβ species in 3D cultures with heterogeneous FAD hNPCs[21,22]. However, it was not possible to dissect the impact of Aβ42/40 ratio versus total Aβ42 or Aβ40 levels on Aβ aggregation in 3D gels due to the heterogeneous nature of these FAD hNPCs. Here, we employed clonal FAD hNPCs producing comparable total Aβ levels with differing Aβ42/40 ratios (Fig. 2c, d) to investigate the impact of the Aβ42/40 ratio on Aβ accumulation/aggregation. As expected, clonal FAD hNPCs showed robust accumulation of Aβ in 3D gels following differentiation, and Aβ accumulation in clonal FAD hNPCs was much higher than in parental heterogeneous FAD hNPCs (Fig. 3a). We also observed accumulation of SDS-resistant multimeric Aβ species as well as Aβ monomers in 7-week differentiated clonal FAD hNPCs, indicative of the pathogenic accumulation of oligomeric Aβ species in 3D cultures (Fig. 3a).

Accumulation of Aβ38, Aβ40, and Aβ42 in conditioned media and 3D gels was also confirmed by MSD Aβ assay (Fig. 3b, c). Notably, 3D-differentiated FAD hNPCs with high Aβ42/40 ratios (#D4 and #H10) showed robust increases of Aβ42 species within 3D gels as compared to FAD cells with low Aβ42/40 ratios (#A5, #A4H1) (Fig. 3c, e). In the case of ReN-mAP4#E6F4 cells, we observed the highest levels of both Aβ42 and Aβ40 within the 3D gels, despite a relatively low Aβ42/40 ratio. Since ReN-mAP4#E6F4 cells express 10-fold higher Aβ40 and 42 levels than other FAD cells, extremely high levels of Aβs lead to robust accumulation of both Aβ40 and Aβ42. However, levels of detergent (1% sarkosyl)-resistant Aβ correlated with Aβ42/40 ratio, but not levels of total Aβ or Aβ42, without exception (Fig. 3f). Thus, ReN-mGAP10#D4 and ReN-mGAP2#H10 clonal lines showed elevated levels of detergent-insoluble Aβ species in 3D gel systems.

Importantly, ReN-mAP4#E6F4 cells displayed moderate accumulation of detergent-resistant Aβ despite the high accumulation of both Aβ40 and Aβ42 in 3D gels (Fig. 3f). We also observe that aggregated Aβ species in FAD hNPCs with high Aβ42/40 ratios formed Aβ plaque-like structures (Fig. 3g and Supplementary Fig. 6) and fibrillar structures within 3D gels that were detected by anti-Aβ 3D6 antibody-conjugated gold nanoparticles (Fig. 3h). Congo red immunofluorescence assay also confirmed the robust increase of aggregated amyloid-like structures in 3D-differentiated FAD hNPCs with high Aβ42/40 ratios (Supplementary Fig. 7). These results demonstrate that Aβ42/40 ratio, but not total Aβ levels (either Aβ40 or Aβ42), determines Aβ aggregation in clonal 3D human neural cell culture models of AD.

**Tau pathology correlates with high Aβ42/40 ratio.** We next investigated the impact of total Aβ versus Aβ42/40 ratio on tau pathology using the 3D-differentiated clonal FAD hNPCs (Fig. 4 and Supplementary Fig. 8). Pathogenic p-tau species can be found in human AD neurons[13–16,21,22,25–28]. Our 3D AD human neuronal cell culture models showed not only robust accumulation of p-tau in neurons but also abnormal detergent-insoluble fibrillar tau aggregates similar to NFT in neurites and cell bodies[21,22]. We conducted immunofluorescence confocal microscopy to assess the accumulation of p-tau in clonal FAD hNPCs that were 3D-differentiated for 6 weeks. We observed dramatic increases in pathogenic p-tau levels, detected by PHF1 (pSer396/404) and AT100 (pThr202/Ser204) anti-p-tau antibodies in 3D-differentiated FAD hNPCs with high Aβ42/40 ratios (Fig. 4a and Supplementary Fig. 8a). In addition, Alz50, an antibody

against aggregated tau species, also revealed that the increased level of aggregated tau was correlated with Aβ42/40 ratio (Supplementary Fig. 9). ELISA analysis showed the accumulation of pathogenic p-tau (AT270, pThr181) in 3D-differentiated FAD hNPCs (Fig. 4b). After normalizing p-tau levels with a neuronal marker, Tuj1, we confirmed a positive correlation between the Aβ42/40 ratio and neuronal p-tau accumulation (Fig. 4c). Levels of detergent-insoluble p-tau and total tau were particularly elevated in 3D-differentiated FAD cells with high Aβ42/40 ratio. Importantly, elevated p-tau and total tau were not clearly observed in cells with low Aβ42/40 levels, e.g. ReN-mAP4#E6F4, even though they contained 10-fold higher levels of Aβ42 as compared to other FAD cells (Fig. 4d). Immuno-EM analysis revealed the presence of mature tau fibrils with straight and paired-helical filament structures in 3D-differentiated FAD hNPCs with high Aβ42/40 ratio (Fig. 4e and Supplementary Fig. 10). To re-confirm that Aβ42/40 ratio regulates tau pathology, we employed pharmacological approach by treating BPN-15606, a γ-secretase modulator, which has been reported to reduce Aβ42/40 ratio in AD cells[29]. We found that BPN-15606 treatments effectively reduced p-tau accumulation in the ReN-mGAP2#H10, a cell line exhibiting the highest level of Aβ42/40 ratio and p-tau accumulation (Figs. 2d, 3d and Supplementary Fig. 8d, e). These results demonstrate that the high Aβ42/40 ratio (not simply high levels of Aβ42) is critical for inducing robust tau pathology, including detergent-insoluble filamentous tau aggregates, in clonal 3D human neural cell culture model of AD.

**Increased cell death correlates with high Aβ42/40 ratio.** Aβ42/40 ratio correlated with p-tau accumulation following normalizing with Tuj1 (Fig. 4c). However, unnormalized p-tau levels were relatively decreased in ReN-mGAP2#H10 cells even though these cells displayed the highest Aβ42/40 ratio in this study (Fig. 4b). To explain this discrepancy, we hypothesized that neuronal death is markedly increased in cells with high Aβ42/40 ratio. To test this, we first performed an unbiased DAPI counting assay in 3D-differentiated control (ReN-G2#B2) and FAD hNPCs with a high Aβ42/40 ratio (ReN-mGAP10#D4 and ReN-mGAP2#H10). Cell numbers in 3D-differentiated ReN-mGAP10#D4 and ReN-mGAP2#H10 cells were significantly decreased as compared to the ReN-G2#B2 cells (Supplementary Fig. 11a). Next, we measured cell death by immunostaining cells with antibodies against cleaved (active) caspase-3. We observed a robust increase in cleaved caspase-3 signals in 3D-differentiated FAD hNPCs with high Aβ42/40 ratio as compared to control (Supplementary Fig. 11b, c). ReN-mGAP2#H10 cells, with the highest Aβ42/40 ratio in this study, also exhibited the highest immunofluorescent signals for active caspase-3 (Supplementary Fig. 11b, c). In addition to increased cell death in 3D-differentiated FAD hNPCs, we observed dramatic decreases in neurite networks in cells with high Aβ42/40 ratios (Supplementary Fig. 11c), which is consistent with neuronal cell death in the 3D FAD models. These results suggest that high Aβ42/40 ratio is correlated with cellular death in 3D-differntiated FAD hNPCs.

**APP TMD mutations modulate Aβ42/40 ratios without PS1ΔE9.** To investigate the direct impact of Aβ42/40 ratio on tau pathology, we generated multiple clonal hNPCs with APP transmembrane domain (TMD) mutations, which achieve extremely high (APP TMD I45F) or low (I47F) Aβ42/40 ratio in the absence of mutant PS1 overexpression (Fig. 5a). We took advantage of a recent finding regarding the sequential PS1/γ-secretases-mediated cleavage of the APP TMD[30]. This study shows that the APP TMD mutation, I45F, dramatically increases the Aβ42/40 ratio by blocking the Aβ49-46-43-40 cleavage

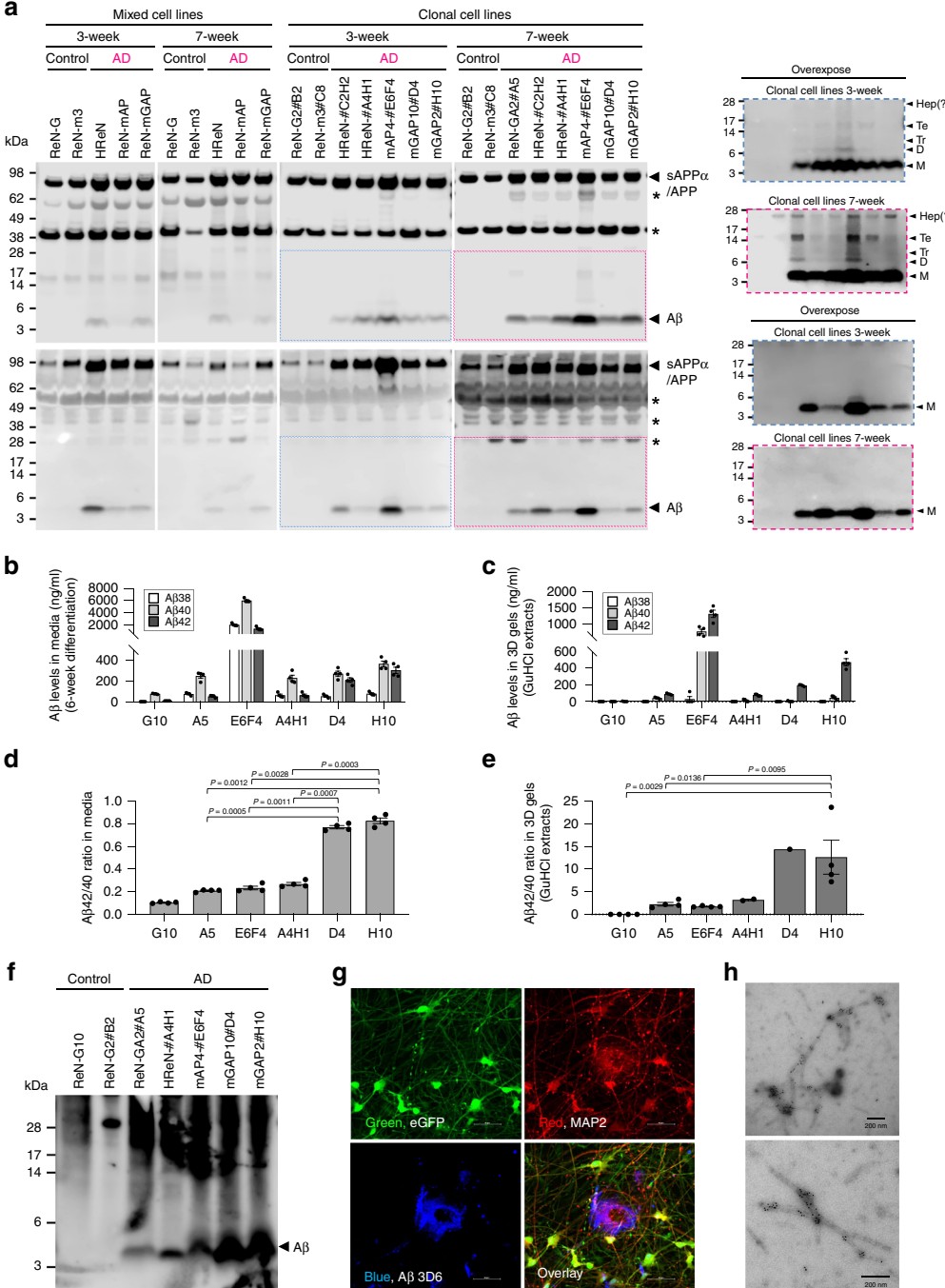

**Fig. 3 Clonal FAD hNPCs with high Aβ42/40 ratio showed robust accumulation of detergent-resistant Aβ42 aggregates. a** Comparison of accumulated Aβ levels between differentiated mixed and clonal AD hNPC lines in 3D gels (top panel) and media (bottom panel). Mixed and clonal hNPCs (1.2 × 10[6]) were 3D-differentiated in 24-well plates for 3 and 7 weeks. SDS-soluble protein lysates were prepared from differentiated 3D cultures. The levels of Aβ peptides in both 3D gels and media were measured by Western blot analysis using anti-Aβ antibody (6E10). The overexposed images, in dashed line rectangles (blue: 3 weeks, red: 7 weeks), are presented on the right. Asterisk represents unidentified bands which might be partially cleaved APP fragments, large Aβ, or nonspecific bands detected by 6E10 antibody. **b–e** The clonal control and FAD hNPCs (3 × 10[5]) were 3D-differentiated in 96-well plates for 6 weeks. Concentrations of Aβ species (Aβ38, Aβ40 and Aβ42) in media (**b**) and in 3D gels (**c**) were analyzed by MSD Aβ assay. Aβ42/40 ratios based on the Aβ concentrations in media (**d**) and in 3D gels (**e**) were presented. Protein extracts from 3D-differentiated clonal hNPCs were prepared using 5 M guanidine hydrochloride (GuHCl). All data are expressed as mean ± SEM of four independent repeats (black dots). Statistical significances were determined by one-way ANOVA with Tukey's multiple comparisons test. Aβ values below the detection limit of the MSD Aβ assay were excluded from calculation of Aβ42/40 ratio. Graphs represent only data validated by MSD Aβ assay. **f** Clonal hNPCs (8 × 10[6]) were 3D-differentiated in 6-well plates for 6 weeks. Samples were harvested and dissolved in 1% sarkosyl lysis buffer. Sarkosyl-insoluble fractions reveal aggregated Aβ species. **g** ReN-mGAP10#D4 cells were differentiated in 3D cultures for 6 weeks. After fixing with 4% paraformaldehyde, cells were immunostained with antibodies against Aβ (3D6) and MAP2. The extracellular Aβ deposition and MAP2-positive cells were imaged by confocal microscopy. **h** Representative electron micrographs show Aβ fibrils. Detergent-insoluble fractions were purified from 3D-differentiated ReN-mGAP10#D4 for 9 weeks. Aβ fibrils were labeled with anti-Aβ antibody (3D6)-conjugated gold nanoparticles.

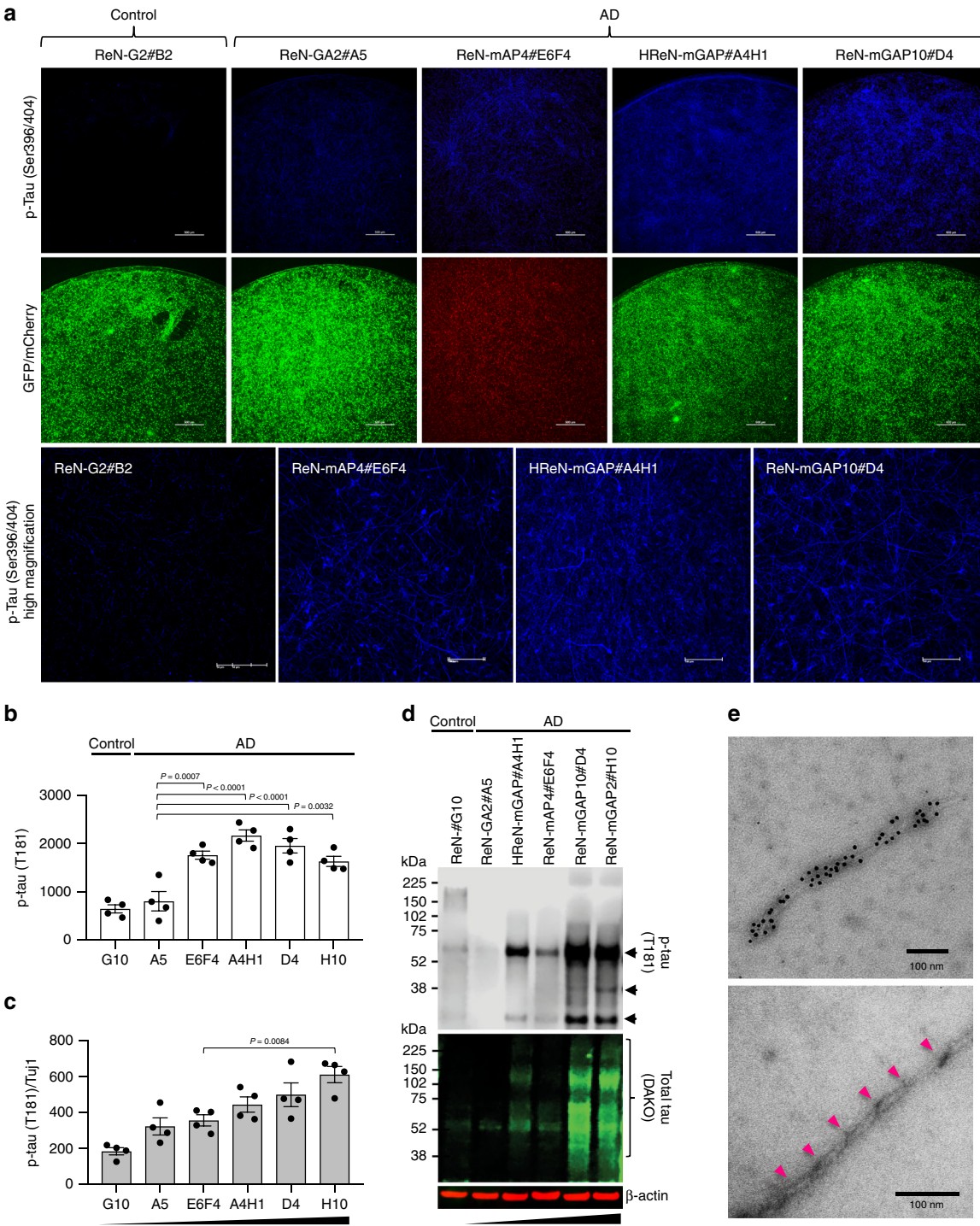

**Fig. 4 Aβ42/40 ratio regulates tau pathology in 3D-differentiated FAD hNPCs. a** Clonal control and FAD hNPCs ($3 \times 10^5$) were 3D-differentiated in 96-well flat-bottom plates for 6 weeks. After fixing with 4% paraformaldehyde solution, cells were immunostained with antibody against p-tau (PHF1; Ser396/404). Enlarged images of PHF1-positive cells in ReN-G2#B2, ReN-mAP4#E6F4, HReN-mGAP#A4H1, and ReN-mGAP10#D4 are presented. Scale bars represent 500 μm. **b** Clonal control and FAD hNPCs ($3 \times 10^5$) were 3D-differentiated in 96-well plates for 6 weeks. Total proteins were extracted by dissolving in 5 M GuHCl lysis buffer. Levels of p-tau (T181) were measured by ELISA assay. **c** Levels of p-tau (T181) in 3D-differentiated clonal hNPCs were normalized by Tuj1. All data are expressed as mean ± SEM of four independent repeats (black dots). Statistical significances were determined by one-way ANOVA with Tukey's multiple comparisons test. **d** Cells ($1.2 \times 10^6$) were 3D-differentiated in 24-well plates for 6 weeks. Samples were harvested and lysed in 1% sarkosyl lysis buffer. Sarkosyl-insoluble proteins were fractioned by ultracentrifugation. Western blot analysis was performed to identify p-tau (T181) (top panel) and total tau (bottom panel) in the clonal hNPCs. For the loading control, β-actin in the sarkosyl-soluble fraction was used. **e** Representative electron micrographs show tau fibrils. Sarkosyl-insoluble fractions were purified from a 3D culture of 9-week differentiated ReN-mGAP10#D4. The tau fibrils were labeled with anti-tau antibody (T46)-conjugated gold nanoparticles (top panel). Scale bars represent 100 μm. The helical structures in tau fibrils are presented (bottom panel).

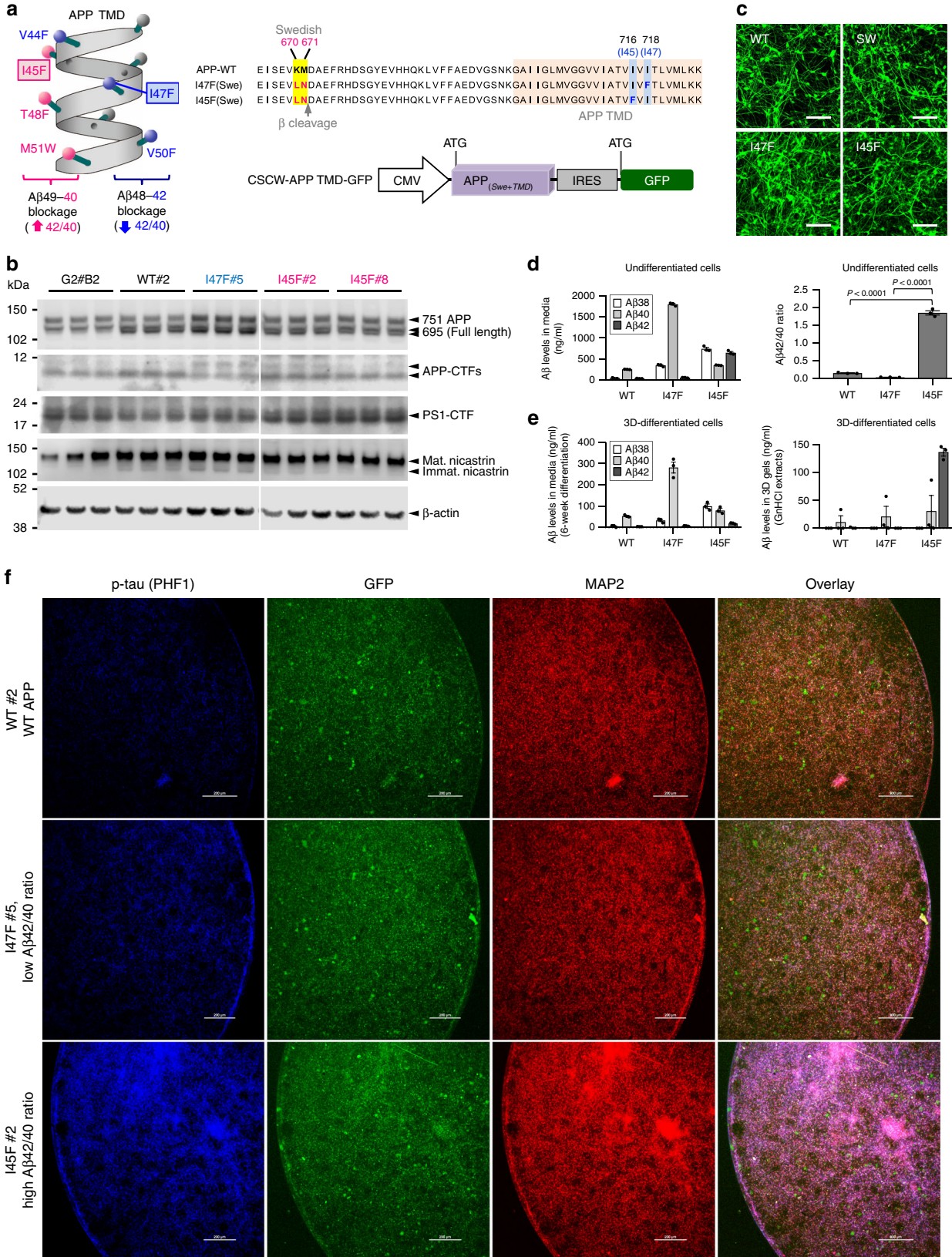

pathway. Conversely, the I47F APP TMD mutation blocks the APP Aβ48-45-42 cleavage pathway, making the lower Aβ42/40 ratio[30]. We selected clonal hNPCs expressing APP WT, APP TMD I47F, and APP TMD I45F with similar levels of APP and APP-CTFs (Fig. 5b). The expression levels of PS1-CTF and

nicastrin in the APP TMD mutant cell lines were also comparable to the ReN-G2#B2 cell line, suggesting that the function of the γ-secretase complex is not adversely impacted by the expression of APP TMD mutant constructs (Fig. 5b). Clonal APP TMD cells, including control APP WT and APP Swedish (SW) alone, showed

**Fig. 5 High Aβ42/40 ratio robustly increases tau pathology in 3D-differentiated hNPCs with APP TMD mutations. a** Schematic illustration of the TMD region in APP protein (left panel). The number assigned to each amino acid along the α-helical structure represents the length of Aβ peptide generated by γ-secretase cleavage. The substitution of amino acids denoted by red spheres with bulky amino acids, phenylalanine (F) or tryptophan (W), favors generation of Aβ42 peptides (high Aβ42/40 ratio) whereas the replacement of amino acids denoted by blue spheres with Fs shifted APP cleavage toward the Aβ40 pathway (low Aβ42/40 ratio). Aligned amino acid sequences with mutations and a schematic illustration of lentiviral DNA construct used for generating hNPC APP TMD mutant cells are presented (right panel). **b** Indicated cells ($2 \times 10^6$) were 2D-plated and grown in 6-well plates for 48 h. Levels of full-length APP, APP-CTF (C83 and C99), PS1-CTF and nicastrin were monitored by Western blot analysis. **c** The representative confocal images for 3-week differentiated clonal hNPCs harboring wildtype APP (WT), APP^Swedish (SW), I47F, and I45F are presented. **d** The Aβ levels in media from undifferentiated hNPCs harboring WT APP or APP TMD mutations (left) and the calculated Aβ42/40 ratio (right). All values were combined to express as mean ± SEM of three independent repeats (black dots). Statistical significances were determined by one-way ANOVA with Tukey's multiple comparisons test. **e** The Aβ levels in 5-week differentiated media (left) and 3D gels (right). The Aβ38, Aβ40 and Aβ42 levels were measured by MSD Aβ assay. Note: Aβ values below the detection limit for MSD Aβ assay were excluded from calculations of Aβ42/40 ratio. Graphs represent only data validated by MSD Aβ assay. **f** Robust increase of p-tau in I45F cells compared to I47F cells. Cells ($3 \times 10^5$ per well) were plated in 3D-thin-layer cultures in 96-well plates and differentiated for 6 weeks. After fixing with 4% paraformaldehyde, cells were stained with antibodies against p-tau (PHF1) and MAP2. Scale bars represent 200 μm.

similar morphology to differentiated clonal FAD hNPCs, when differentiated into neuronal cells under 3D culture conditions (Fig 5c).

**High Aβ42/40 ratio drives Aβ and p-tau accumulation.** APP TMD I47F hNPCs mainly expressed Aβ40 with extremely low Aβ42 levels, leading to an Aβ42/40 ratio lower than that in WT cells. In contrast, I45F cells predominantly produced Aβ42 as compared to Aβ40, making the Aβ42/40 ratio ~57-fold higher than that in I47F cells (Fig. 5d). Following 3D differentiation, we observed robust increases in levels of insoluble Aβ42 species (5 M GuHCl extracted) in I45F cells and undetectable levels in 3D-differentiated I47F hNPCs (Fig. 5e), consistent with our findings in 3D-differentiated hNPCs with APP and PS1 overexpression (Fig. 3). Interestingly, Aβ42 levels were very low in conditioned media collected from 3D-differentiated I45F cells, possibly due to a preferential aggregation of Aβ42 in 3D gels (Fig. 5e). Indeed, we performed confocal immunofluorescence analysis to detect aggregated Aβ species in the gel and observed that the number of Aβ aggregation was dramatically increased in 3D gels with differentiated APP TMD I45F cells but less number of 3D6-positive signals were detected in control (ReN-G10) or APP TMD I47F cells (Supplementary Fig. 13). These results confirmed that the Aβ42/40 ratio regulates Aβ42 accumulation/aggregation in 3D gels.

To analyze tau pathology in 3D-differentiated APP TMD hNPCs, we performed confocal immunofluorescence analysis of pathogenic p-tau levels in 5-week or 7-week 3D-differentiated WT, I47F, and I45F cells (Fig. 5f and Supplementary Fig. 14). The level of p-tau was drastically elevated in the I45F cells (high Aβ42/40 ratio) as compared to the I47F cells (low Aβ42/ Aβ40 ratio), suggesting that upregulation of p-tau levels are determined by the Aβ42/40 ratio, but not induced by changes in neuronal cell number or neurite networks.

**Aβ42/40 ratio regulates tau pathology in naïve hNPCs.** Next, we tested whether the effects of APP TMD mutant cells and associated increase in Aβ42/40 ratio on tau pathology could be due to overexpression of human APP. Prior studies have suggested multiple roles for full-length APP and its cleavage products (other than Aβ) in cellular functions[9,31–37]. Therefore, we attempted to validate the impact of Aβ42/40 ratio on tau pathology in naïve hNPCs lacking FAD mutations, using a 3D non-cell autonomous culture model (Fig. 6a). For this purpose, we prepared 3D-thick-layer cultures with naïve hNPCs (ReNcell VM) in cell culture inserts. They were co-cultured with cells overexpressing wildtype APP (WT) or APP TMD mutations

(I45F or I47F), which were seeded at the bottom of multi-well plates in 2D (Fig. 6a).

In this 3D co-culture system, there was no direct contact between 3D naïve hNPCs and 2D APP TMD mutant cells. After 12 weeks of co-differentiation, we detected a dramatic accumulation of pathogenic p-tau (PHF, pThr398/S404) in 3D naïve hNPCs co-cultured with I45F APP TMD cells. However, there was no significant increase in the cells with either control WT or I47F cells (Fig. 6b, d). More importantly, detergent-insoluble (1% sarkosyl) aggregated p-tau levels were dramatically increased in 3D naïve cells with I45F cells, but not with WT or I47F cells (Fig. 6c, e). Detergent-insoluble tau levels were also increased in 3D naïve hNPCs co-cultured with I45F mutant cells even though they did not reach statistical significance (Fig. 6c, e). Taken together our results strongly demonstrate that the Aβ42/40 ratio directly regulates accumulation of aggregated tau species in naïve hNPCs.

**Discussion**

In this study, we showed that Aβ42/40 ratio, but not total Aβ levels, regulates tau pathology in human neural cells, using multiple 3D human neural cell culture models with single cell-derived clonal AD stem cells. Previously, we showed that excess accumulation of Aβ induces robust tau pathology, including sliver-stained and detergent-insoluble filamentous tau species, in 3D human neural cell cultures system[21]. However, due to the heterogeneity of the cells used in our previous studies, it was difficult to directly address the impact of Aβ42/40 ratio versus total Aβ levels on advanced tau pathology and other Aβ42-driven pathological changes. To overcome this limitation, we generated clonal FAD hNPCs derived from heterogeneous parental FAD hNPCs (Fig. 1). These clonal 3D AD cell cultures provide homogeneous conditions to investigate the role of Aβ42/40 ratio in regulating advanced tau pathology, including abnormal accumulation/aggregation of p-tau in human neuronal cells (Figs. 3 and 4).

We also employed 3D cell culture models with APP TMD mutant hNPCs (I45F and I47F) to confirm that the p-tau accumulation/aggregation was not induced by mutant PS1 overexpression but specifically by the elevated Aβ42/40 ratio engendered by PS1 FAD mutations (Fig. 5). In addition, to exclude autonomous effects of the overexpressed FAD mutations on neuronal tau phosphorylation, we employed 3D non-cell autonomous cell culture system. We showed that elevated Aβ42/40 ratio strongly induces advanced tau pathology in naïve human neural cells lacking FAD mutations (Fig. 6). Together, our results provide clear evidence connecting elevated Aβ42/40 ratio to tau pathology and NFTs in human neurons.

To dissect a potential contribution of presenilin FAD mutations on AD pathology in our 3D AD cellular models, we also

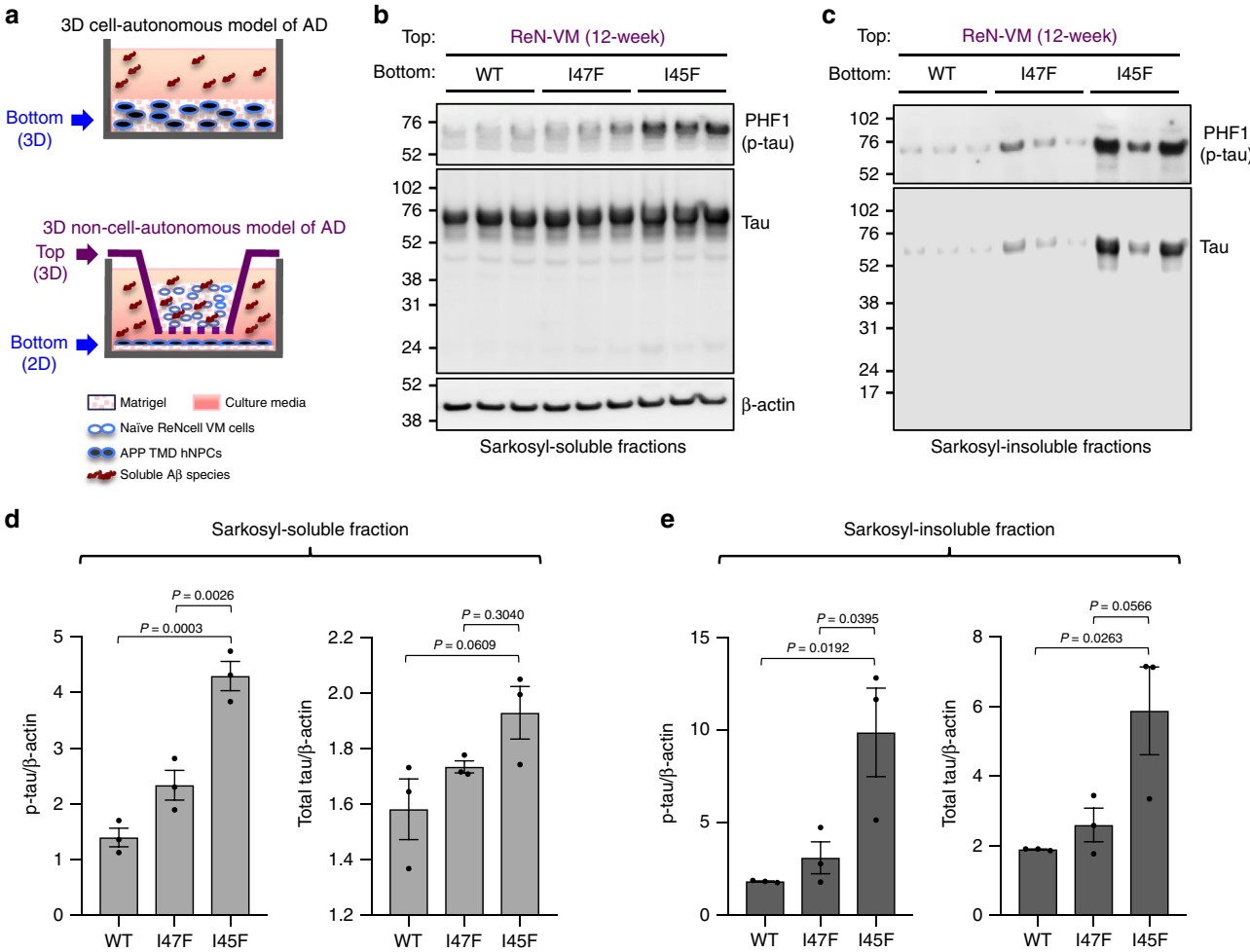

**Fig. 6 Aβ42/40 ratio regulates tau pathology in naïve hNPCs without FAD mutations. a** Schematic illustrations of conventional 3D cell autonomous (top panel) and 3D non-cell autonomous (bottom panel) culture models of AD. Both thick-layer 3D cultures of naïve hNPCs (ReNcell VM) ($2.5 \times 10^6$) plated in cell culture inserts (labeled with top) and 2D-plated APP TMD mutant cells ($5 \times 10^5$), APP-WT, I47F, and I45F, respectively, in 24-well plates (labeled with bottom) are differentiated for a week separately. To prepare the non-cell autonomous 3D cultures, one-week differentiated cells are combined together in the same wells of 24-well plates, which differentiate for additional 12 weeks. **b**, **c** Naïve hNPCs co-cultured with APP TMD mutant cells for 12 weeks were dissolved in 1% sarkosyl lysis buffer to prepare sarkosyl-soluble (**b**) and sarkosyl-insoluble (**c**) fractions as described in Methods. The levels of p-tau (PHF1) and total tau were examined by Western blot analysis. β-actin was used as a loading control. **d**, **e** Relative levels of p-tau and total tau in sarkosyl-soluble (**d**) and sarkosyl-insoluble (**e**) were quantified using Fiji software. All values were normalized with β-actin levels. The results were combined to express as mean ± SEM of three independent repeats (black dots). Statistical significances were determined by one-way ANOVA with Tukey's multiple comparisons test.

generated human neural stem cells that overexpress presenilin 1 with ΔE9 familial mutation alone (ReN-m-PS1ΔE9), and compared the impact of PS1ΔE9 on tau pathology in our 3D cellular model. Previous studies suggest that presenilin FAD mutations induce multiple cellular deficits including intracellular calcium homeostasis[38]. We observed that overexpression of PS1ΔE9 alone did not alter phosphorylated or total tau levels as compared to control cells after 3D differentiation (Supplementary Fig. 12). Since we have showed robust accumulation of p-tau in cells expressing both APPSL and PS1ΔE9, this result strongly supports that accumulation of pathogenic Aβ species is mainly responsible for tau pathology in our 3D human neural cell culture models of AD.

Previous genetic and biochemical studies strongly suggest that accumulation of pathogenic Aβ42 (and the elevated Aβ42/40 ratio) plays a primary role in triggering pathological cascades in AD[1,2,39,40]. However, a direct causal relationship between the increased Aβ42/40 ratio and tau pathology (including filamentous tau aggregation) has never been proven in human neurons before.

Human iPSC-derived neurons that contain strong familial AD mutations could not recapitulate Aβ42-driven pathological changes. Instead, these studies suggest that the accumulation of APP C-terminal fragments (APP-CTFs), not pathogenic Aβ42, is responsible for endosomal trafficking deficits and altered tau metabolism (partially)[14,41]. In our opinion, these neuronal models do not achieve enough pathogenic Aβ42 accumulation, which is comparable to AD brains, and therefore, may not be able to comprehensively recapitulate AD pathological cascades that are triggered by robust accumulation of pathogenic Aβ species in AD brains. Thus, our results provide an important mechanical insight into molecular mechanism of how tau pathology can be triggered in human AD neurons.

Previous studies showed that there is a dramatic accumulation of Aβ42 species in both FAD and SAD[1]. According to Wang et al., Aβ42/40 ratios are increased by ~3 fold both in TBS soluble and formic acid fractions in the brain of AD patients as compare to the aged control[42]. Studies also show that there are significant decreases in Aβ42/40 ratio in brain CSF at early stage of SAD,

which has become a standard biomarker for AD diagnosis[43–46]. The dramatic decreases in the CSF Aβ42/40 ratio in AD patients are possibly caused by selective accumulation/aggregation of Aβ42 species in brain parenchyma, supporting the increase of Aβ42/40 ratio in brains of SAD[45]. Therefore, our conclusion that Aβ42/40 ratio drives p-tau pathology can be applicable to both FAD and SAD. Since our non-cell autonomous model of AD recapitulated conditions with high Aβ42/40 ratio using APP TMD mutant hNPCs (Fig. 6), this model may serve for studying pathogenic mechanism of SAD.

The Aβ cascade hypothesis suggests that pathogenic Aβ oligomers play a central role in triggering pathogenic cascade in the brain of AD patients[1,47]. High Aβ42/40 ratio would lead to produce more toxic Aβ oligomeric structures. In vitro studies showed that Aβ40 and Aβ42 form distinctive oligomeric structures[47]. Interestingly, Aβ42/Aβ40 mixture rapidly formed small spherical oligomers, which were more toxic than oligomeric preparations composed of either Aβ40 or Aβ42 alone[48–50]. Studies also showed that Aβ dimers and protofibrils induce synaptotoxicity[51,52]. Our results with 3D non-cell autonomous models (Fig. 6) also supports the notion that soluble species, generated in the condition of a high Aβ42/40 ratio, are enough to induce robust tau pathology in 3D-differentiated naïve neurons. As an alternative hypothesis, Aβ40 may independently play a protective role, which can counteract Aβ42 toxicity in our model. The high Aβ42/40 ratio can also increase levels of insoluble Aβ fibrils and aggregates in 3D gels and this may directly trigger tau pathology in neurons. Further studies will be needed to clarify the roles of Aβ oligomers, protofibrils, other fibrillar forms and possibly other Aβ isoforms including Aβ43, in inducing tau pathology in the 3D human neural cell culture models[52,53].

Human iPSC-derived neurons from AD patients provide a physiological tool to study the pathogenic impact of Aβ42, Aβ40 and Aβ42/40 ratio[14,15,25,27]. Human neurons harboring not all, but select familial AD mutations, also display increased p-tau and total tau levels, suggesting an early stage of tau pathology[14,16,25,54,55]. However, these neurons do not show robust aggregation of Aβ (Aβ plaque) or tau (NFTs). In addition to the low levels of Aβ42 in these iPSC-derived cell models, as compared to the brain of AD patients, these neurons do not express substantial levels of adult 4-repeat (4R) tau isoform even after extended 2D differentiation without *MAPT* mutations that accelerate the splicing of 4R tau isoform[56,57]. In Supplementary Fig. 5d, we re-confirmed the substantial expression of exon 10 which is necessary to produce 4R tau isoform in our clonal 3D AD cellular models[21]. This result may explain the discrepancy between previous AD iPSC models and our current 3D models. Furthermore, these studies have not shown any correlation between the Aβ42/40 ratio and p-tau accumulation with iPSC-derived neurons from AD patients[13,14,16,25,55,58]. Most neurons harboring PS1/2 mutations exhibit an increased Aβ42/40 ratio but no changes in p-tau level[14]. Human iPSC-derived neurons harboring APP duplication or select APP mutation (APP V717F) show p-tau accumulation, but only after 3–6 months of differentiation[14,25]. APP duplication increases general Aβ levels, but not the Aβ42/40 ratio, while APP V717I mutation increases the Aβ42/40 ratio. As we mentioned above, studies also show that partial tau metabolism can be altered by endosomal trafficking deficit induced by APP-CTF accumulation[14,41]. To date, therefore, it is not clear whether increased Aβ levels or Aβ42/40 ratio regulate tau pathology in human neurons. However, this study provides direct evidence that elevated the Aβ42/40 ratio drives accumulation and aggregation of pathogenic tau species in human neurons. Pathological studies suggest that cognitive decline and neurodegeneration are tightly connected to tau pathology, but not to Aβ plaques[59,60]. We also observed elevated

active caspase-3 levels and neurite network degeneration in 3D-differentiated FAD hNPCs with high Aβ42/40 ratio (Supplementary Fig. 11b, c), which is a sign of neuronal death in FAD cells with high Aβ42/40 ratio, as well as advanced tau pathology. Moreover, we have shown that the addition of human microglia to our 3D FAD hNPCs model robustly increased neuroinflammation and neurodegeneration[61]. Further studies will be needed to clarify whether the elevated cell death in these models is mediated by Aβ42 levels or Aβ42/40 ratio, or by tau pathology triggered by high Aβ42/40 ratio.

Recently, multiple clinical trials of therapies aimed at blocking Aβ accumulation have failed[62–64]. These failures suggest that anti-Aβ treatments are administered too late to reverse Aβ42-driven pathogenic changes in symptomatic AD, which may have been started two decades before treatments[65]. In line with this idea, early prevention trials targeting Aβ accumulation are currently ongoing. An alternative approach is to focus on blocking specific pathogenic cascades triggered by Aβ42 in human neuronal cells. Together with current AD mouse models that focus on Aβ accumulation and subsequence cognitive decline, our clonal 3D human AD cellular models can provide a unique experimental platform to explore how Aβ42-specific pathogenic cascades, e.g. owing to elevated Aβ42/40 ratios lead to NFTs and subsequent neurodegeneration. These pathways may provide novel drug targets that can reduce Aβ-driven pathological changes in AD patients. We also showed that the clonal FAD hNPCs could maintain high Aβ42 and Aβ42/40 ratios for as many as 35 passages (Supplemental Fig. 15), demonstrating that this system can provide a consistent and replicable drug testing environment to be applicable to large scale AD drug screening.

Collectively, our current findings suggest that reducing Aβ42/40 ratio is more important than generally decreasing total Aβ levels. Recent clinical trials targeting γ-secretase and β-secretase, both of which were designed to generally decrease total Aβ levels including Aβ42 and Aβ40, have failed. Our clonal 3D models clearly show that the Aβ42/40 ratio, not total Aβ levels, governs advanced tau pathology and Aβ42-driven pathological pathways in human neurons and that going forward it will be important to develop alternative therapeutic approaches that selectively reduce Aβ42/40 ratio.

## Methods

**Cell culture, reagents, and antibodies**. ReNcell VM expansion and differentiation media were prepared as previously described[22]. Control (non-FAD) and FAD hNPCs were expanded in ReNcell expansion media containing DMEM/F-12 media (1132-033, Life Technologies) supplemented with 2 μg/ml heparin (07980, StemCell Technologies), 2% (v/v) B27 neural supplement (17504-044, Life Technologies), 20 ng/ml EGF (E9644, Sigma-Aldrich), 20 ng/ml bFGF (03-002, Stemgent, Cambridge), 100 U penicillin/100 μg streptomycin (17-602E, Lonza), 125 μg amphotericin-B solution (30-003-CF, Lonza), and plated on a BD Matrigel (354234, BD Biosciences)-coated T25 or T75 cell culture flasks (BD Biosciences). Cells were incubated in a humidified atmosphere containing 5% $CO_2$ at 37 °C. Differentiation media was prepared using the above recipe without EGF and bFGF. N-lauroylsarcosine (sarkosyl) (L5777), guanidine hydrochloride (GuHCl) (G3272) and phenylmethylsulfonyl fluoride (PMSF) (P7626) were purchased from Sigma-Aldrich. Rabbit polyclonal β-Tubulin III (Tuj1) antibody (T2200) and HRP-conjugated rabbit polyclonal β-actin antibody (A3854) were purchased from Sigma-Aldrich; mouse monoclonal Aβ antibody (6E10, SIG-39320) from BioLegend; mouse monoclonal Phospho-Tau T181 antibody (AT270, ENMN1050) and mouse monoclonal tau antibody (BT2, MN1010) from Thermo Fisher Scientific; rabbit polyclonal antibodies against tau (A002401-2) and GFAP (Z0334) from DAKO (Agilent Technologies); mouse monoclonal β-Tubulin III (Tuj1) antibody (TU-20, 4420), mouse monoclonal tau antibody (Tau46, 4019) and rabbit monoclonal cleaved caspase-3 antibody (5A1E, 9664) from Cell Signaling Technology; chicken polyclonal MAP2 antibody (ab5543) from EMD Millipore; PHF1, Alz50 and MC1 antibodies were kindly provided by Dr. Peter Davies (The Feinstein Institute for Medical Research, Donald and Barbara Zucker School of Medicine at Hofstra/Northwell, Manhasset, NY, USA) and 3D6 anti-Aβ antibody was a gift from Eli Lilly (Cambridge, MA, USA). A soluble γ-secretase modulator, BPN-15606 was provided by Dr. Steven L. Wagner (Department of Neurosciences, University of California, San Diego, La Jolla, CA, USA)

**3D cell culture**. The 3D culture was performed as described with some modifications[22]. The hNPCs were resuspended in ice-cold differentiation media and kept on ice. For thin-layer 3D cultures, BD Matrigel stock solution was added to ice-cold cell suspensions (1:10 dilution rate) to a final concentration of $2 \times 10^6$ cells/ml in 10% (v/v) Matrigel. Cells were added to ice-cold differentiation media in 10% (v/v) Matrigel. The cell/Matrigel mixtures were immediately transferred into multi-well plate using pre-chilled pipettes. The plating volume was 150 µl/well for 96-well plate, 600 µl/well for 24-well plate and 4 ml/well for 6-well plate. The following day, fresh differentiation media was added.

The 3D cultures were differentiated for 5–12 weeks depending on the experiment. Media was changed every 3 or 4 days. The cells were either fixed with 4% paraformaldehyde for immunostaining or harvested for Western blot analysis and RNAseq analysis. For biochemical analysis requiring higher amount of cell mass, we used slightly modified 3D culture conditions. In this "expansion-differentiation" protocol, the cells were resuspended in ice-cold expansion media in 10% (v/v) Matrigel, plated and expanded for 24 h. Then, expansion media was switched to differentiation media for the rest of 3D differentiation.

A non-cell autonomous cell culture system was designed to differentiate 3D-thick-layer-cultured naïve ReNcell VM cells and 2D-cultured APP TMD mutant cells together in a single culture vessel. For preparing 3D-thick-layer-culture, naïve ReNcell VM cells ($2 \times 10^7$ cells/ml) were suspended in ice-cold differentiation media and diluted with an equal volume of BD Matrigel (1:2 dilution rate). After vortexing for 20 s, 250 µl of the cell/Matrigel mixture was immediately transferred into tissue culture inserts (662630, ThinCerts, 3.0 µm pore size, Greiner Bio-One) which were placed in 24-well plates and incubated for 1 h at 37 °C to harden the gels. Then, 1 ml of pre-warmed differentiation media was added to the bottom of the plates. After incubating for an additional 24 h, 0.3 ml of pre-warmed differentiation media was added into the inserts. Clonal hNPCs overexpressing wildtype APP (WT) or APP TMD mutant cells (I47F or I45F) ($5 \times 10^5$) were seeded in 2D on bottom of 24-well plates. After one-week of differentiation, the inserts with 3D-cultured naïve ReNcell VM cells were transferred to the 24-well plates where the APP TMD mutant cells were grown. The cells were differentiated together for 12 weeks as media was changed every 3-4 days.

**FACS-assisted single cell isolation**. Parental control (non-FAD) and FAD hNPCs were dissociated by StemPro® Accutase® (A11105-01, Life Technologies) and centrifuged at $500 \times g$ for 2.5 min. Cell pellets were resuspended in ice-cold PBS with 2% KnockOut$^{TM}$ Serum Replacement solution (10828028, Life Technologies) and 2% B27 followed by passed through a cell strainer (352350, 70 µm Nylon, BD Biosciences) to obtain a uniform single cell suspension. After adjusting cells to a density of $2 \times 10^6$ cells/ml, cell suspensions were applied to FACS-assisted single cell sorting using a Laser BD FACSAria Fusion Cell Sorter, BSL2 + (MGH Pathology: Flow, Image and Mass Cytometry Cores, Charlestown, MA, USA). The viability of all cell lines, including controls expressing GFP or mCherry only, was monitored by Forward Scatter Area (FSC-A) versus Side Scatter Area (SSC-A) analysis. The subset of the viable cells was further gated by FSC Height (FSC-H) versus FSC Width (FSC-W) and by SSC-H versus SSC-W dot plots to select a singlet population. Each individually sorted cell was placed into a single well of BD Matrigel-coated 96-well tissue culture plates. After a brief centrifugation step, cells were grown in expansion media as described above. We selected #A5 clone from the ReN-GA line and #D4 and #H10 clones from the ReN-mGAP line for further characterizations. Single cells from the ReN-mAP line were FACS sorted using the above protocol, however, these clones failed to expand in isolation. To assist with expansion and colony formation, we added naïve ReNcell VM cells to the wells with clonal AD cells and expanded them together until we observed visible red fluorescent colonies. Each viable co-cultured colony was FACS sorted again to obtain the mCherry-positive transgenic cells only.

**Quantitative analysis of Aβ levels**. Levels of Aβ38, Aβ40 and Aβ42 in media were simultaneously measured by a multi-array electrochemiluminescence assay kit (K15200E-2, V-PLEX Aβ Peptide Panel 1 (6E10) kit, Meso Scale Diagnostics (MSD)). To quantify Aβ levels in expansion media, cells ($3 \times 10^6$) were seeded in Matrigel-coated 6-well plates in 2D and grown for 24 h. The media was replaced with 1 ml of fresh expansion media. After incubating cells for additional 24 h, conditioned media samples were collected, diluted 1:6 with MSD dilution buffer, and analyzed using the assay kit. To measure Aβ levels from differentiated cell cultures, cells ($3 \times 10^5$) were 3D-differentiated in 96-well plates for 6 weeks. Conditioned media was collected and diluted 1:6, with an exception of conditioned media from ReN-mAP4#E6F4 cells, which were diluted 1:10. To analyze accumulated Aβ levels in 3D gels, total proteins from 6-week differentiated 3D cultures were extracted using 5 M guanidine hydrochloride (GuHCl) lysis buffer followed by analysis of Aβ levels as described above.

**Quantitative analysis of total tau and p-tau levels**. To analyze levels of total tau and p-tau (T181), protein lysates were prepared from 3D-differentiated control and FAD hNPCs ($3 \times 10^5$) plated in 96-well cell culture plates for 6 weeks using 5 M GuHCl lysis buffer. Quantification of tau and p-tau (T181) levels was performed by customized MSD assay based on BT2 tau antibody (Thermo Fisher Scientific) and pThr181 tau antibody (Thermo Fisher Scientific).

**Immunostaining**. 3D-differentiated hNPCs were fixed with 4% paraformaldehyde at room temperature for 24 h. Fixed cells were blocked with 4% skim milk in 1× Tris-buffered Saline (TBS) with 0.1% (v/v) Tween-20 (TBS-T) for an additional 24 h at 4 °C. After washed with 1× TBS-T three times, the cells were permeabilized with a buffer containing 4% goat serum and 0.5% Triton X-100 in TBS-T for 45 min at room temperature. After brief washing with 1× TBS-T, primary antibodies were incubated in buffer containing 50 mM Tris-Cl (pH 7.4), 0.1% Tween-20, 4% donkey serum, 1% BSA, 0.1% gelatin and 0.3 M glycine at 4 °C for 24 h (3D6, 1:400; 6E10, 1:500; PHF1, 1:500; MAP2, 1:500). The cells were washed three times with 1× TBS-T and incubated with Alexa Fluor secondary antibodies for 3.5 h at room temperature with gentle rocking (AlexaFluor 488/568 anti-mouse, -rabbit and -chicken secondary antibodies, 1:400; Cy5 anti-mouse and -rabbit secondary antibodies, 1:400). Fluorescence images were captured by Nikon C2s confocal laser scanning microscope (Nikon Instruments Inc.).

**EM for sarkosyl-insoluble Aβ and tau fibrils**. To perform immunoelectron microscopy, detergent-insoluble samples were placed on formvar-carbon coated Ni grids and allowed to adsorb for 15 min.; residual sample was blotted. For Aβ or tau fibrils, grids with applied sample were placed on drops of 3D6 anti-Aβ antibody (1:20) or on drops of Tau46 anti-tau antibody (1:20, Cell Signaling Technology) and incubated for 1 h at room temperature. After rinsing on drops of PBS, grids were incubated with goat-anti-mouse 10 nm gold (15751, Ted Pella) for 1 h. Grids were then washed with filtered distilled water and stained with 2% phosphotungstic acid or 2% uranyl acetate (Electron Microscopy Sciences) for 2–5 min. Imaging was performed using a JEOLJEM1011 transmission electron microscope at 80 kV. Images were collected using an Advanced Microscopy Techniques digital imaging system with proprietary image capture software.

**RNA purification and RNAseq analysis**. Total RNA was extracted using QIAzol Lysis Reagent (79306, Qiagen) combined with a column-based RNA extraction (Qiagen), according to the manufacturer's protocol. cDNA library preparation, RNA sequencing, quality control and differential gene expression analysis were conducted by the MGH NextGen Sequencing Facility (Simches Research Center, Boston, MA, USA). Sequencing was performed on an Illumina HiSeq 2500 instrument, resulting approximately 30 million 50 bp reads per sample. Sequencing reads were mapped in a splice-aware fashion to the human reference genome (hg19 GRCh37.75) using STAR alignment tools[66]. Read counts over transcripts were calculated using HTSeq[67] based on the Ensemble annotation for GRCh37.75/hg19 assembly. Differential gene expression analysis was conducted using EdgeR[68]. Genes with a false discovery rate (FDR)—adjusted p-value below 0.05 and an absolute log2 (fold change) greater than 1 were classified as differentially expressed.

**ReNcell VM cells with APP TMD mutations**. The lentiviral DNA constructs encoding full-length human APP$^{695}$-K670N/M671L (Swedish) with mutations on the transmembrane domain (TMD) region were generated using a QuikChange Site-Directed Mutagenesis Kit (200523, Agilent Technologies). To generate the CSCW-APP$^{Swe}$-IRES-GFP vector, the London mutation (V717I) in the CSCW-APP$^{Swe/Lon}$-IRES-GFP vector was eliminated using a primer set, 5′-TCA-TAGCGACAGTGATCGTCATCACCTTGGTGATG-3′ and 5′-CATCAC-CAAGGTGATGACGATCACTGTCGCTATGA-3′. The APP TMD mutations, I47F (low Aβ42/40 ratio) and I45F (high Aβ42/40 ratio), were introduced to the CSCW-APP$^{Swe}$-IRES-GFP vector. Recovery of Swedish mutation was performed to generate a lentiviral DNA construct encoding wildtype human APP$^{695}$ protein. The primer sets used for the PCR-based site-directed mutagenesis: I47F, 5′-GTCA-TAGCGACAGTGATCGTCTTCACCTTGGTGATGCTGA-3′ and 5′-TCAG-CATCACCAAGGTGAAGACGATCACTGTCGCTATGAC-3′; I45F, 5′-GTTGTCATAGCGACAGTGTTCGTCATCACCTTGGTGATGC-3 and 5′-GCATCACCAAGGTGATGACGAACACTGTCGCTATGACAAC -3′; APP WT, 5′-GGAGGAGATCTCTGAAGTGAAGATGGATGCAGAATTCCG-3′ and 5′-CGGAATTCTGCATCCCATCTTCACTTCAGAGATCTCCTCC-3′. The mutations on the lentiviral DNA constructs were confirmed by DNA sequencing (MGH Sequencing Core, Charlestown, MA, USA). The viral particles were packaged using Lenti-X$^{TM}$ Packaging Single Shots (VSV-G) kit (631275, Clontech Laboratories). The viral solution ($1 \times 10^6$ TU/ml) was added to 85% confluent proliferating ReNcell VM cells in 6-well plates. After incubating cells for 48 h, the infection process was terminated by washing cells three times with fresh media. The single cell clones expressing APP WT or APP TMD mutants were obtained by the FACS-assisted single cell sorting process as described in Fig. 1a.

**Sarkosyl fractionation from 3D-differentiated hNPCs**. Matrigel embedded differentiated cells were harvested in sarkosyl lysis buffer supplemented with protease and phosphatase inhibitor cocktails (Thermo Fisher Scientific). Lysed samples were homogenized by pipetting, incubated on ice for 30 min, sonicated for 10 min at 4 °C, and centrifuged at $300,000 \times g$ for 30 min at 4 °C. The supernatants containing sarkosyl-soluble fractions were collected. To prepare sarkosyl-insoluble fractions for Western blotting, the pellets were carefully rinsed twice with 100 µl of ice-cold PBS and dissolved in 1× LDS sample buffer (NP0007, Invitrogen) with 2% β-mercaptoethanol (M6250, Sigma-Aldrich). The samples were boiled at 95 °C for 5 min followed by Western blot analysis to detect insoluble tau proteins.

**Aβ extraction from 3D cultures**. Collected 3D cultures were homogenized with ice-cold PBS by pipetting. Samples were centrifuged at $10,000 \times g$ for 10 min at 4 °C. Pellets were dissolved in two-volumes of 2x SDS extraction buffer containing 4% SDS, 100 mM Tris-HCl (pH8.0), 300 mM NaCl, 2 mM EDTA (pH 8.0), 2 mM PMSF, 10 mM PNT and 2% Triton X-100 with protease and phosphatase inhibitor cocktails. After disrupting pellets by pipetting, samples were sonicated for 10 min followed by incubation on ice for additional 30 min. The prepared samples were analyzed with Western blot analysis to detect Aβs accumulated in the 3D gels.

**Congo red staining**. Sarkosyl-insoluble fractions were prepared from 3D-differentiated ReN-G2#B2, ReN-GA2#A5, HReN-mGAP#A4H1 and ReN-mGAP10#D4 cells for 6–9 weeks. Insoluble fractions were resuspended in 1× DPBS (17-512Q, Lonza), spotted and dried on microscope glass slides (12-550-15, Fisher Scientific). Slides were incubated in alkaline 1% Congo red solution containing 0.01% (w/v) NaOH (HT60, Amyloid Stain, Congo Red Kit, Sigma-Aldrich) for 2 h at room temperature and rinsed with double distilled water (DDW) three times for 15 min.

**Statistical analysis**. All statistical analyses were performed using GraphPad Prism v.6 software (Graphpad). Exact values for experimental numbers and p-values, and the types of the statistical test used are reported in the figures and corresponding figure legends. Bars and error bars on the graphs represent mean values and SEM for multiple independent experiments as specified in each legend. Statistical significances were determined by unpaired Student's $t$ test for two groups or one-way ANOVA with Tukey's multiple comparisons test for multiple groups.

**Reporting summary**. Further information on research design is available in the Nature Research Reporting Summary linked to this article.

## Data availability

Raw data for Figs. 1d, e; 2b–d; 3a–f; 4b–d, g; 5b, d, e; 6b–e; Supplementary Figs. 2; 3; 4c–e; 8b, c; 11a, b, 12, 14b and 15 are provided as a Source Data file. All data are available from the corresponding author upon reasonable request.

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

## Acknowledgements

This work is supported by the grants from the Cure Alzheimer's fund (D.Y.K. and R.E.T.), the JPB Foundation (R.E.T.), and National Institute of Health grants: 1RF1AG048080-01 (D.Y.K. and R.E.T.), 5P01AG15379 (D.Y.K. and R.E.T.), 2R01AG014713 (D.Y.K.) and 5R37MH060009 (R.E.T.). We appreciate Drs. Michael Wolfe (University of Kansas) and Dennis J. Selkoe (Brigham and Women's Hospital) for their helpful advice and providing reference APP TMD plasmids. We also appreciate Dr. Peter Davies (Albert Einstein College of Medicine, Bronx, USA) for providing PHF1, Alz50 and MC1 tau antibodies. We would like to thank Ms. Adalis Maisonet and Mr. Ulandt Kim at MGH Next Generation Sequencing Core for the technical assistance. We also appreciate Dr. Wilma Wasco (Massachusetts General Hospital, Boston, USA) for revising the manuscript. Electron microscopy was performed in the Microscopy Core of the Center for Systems Biology/Program in Membrane Biology, which is partially supported by an Inflammatory Bowel Disease Grant DK043351 and a Boston Area Diabetes and Endocrinology Research Center (BADERC) Award DK057521. Cytometric findings reported here were performed in the MGH Department of Pathology Flow and Image Cytometry Research Core, which obtained support from the NIH Shared Instrumentation program with grants 1S10OD012027-01A1, 1S10OD016372-01, 1S10RR020936-01, and 1S10RR023440-01. Lentiviral packaging was performed at MGH Viral Vector Core (supported by NIH/NINDS P30NS04776, PI Dr. Bakhos A. Tannos).

## Author contributions

D.Y.K. and R.E.T. were responsible for experimental design, data analysis and the supervision of this project. S.S.K. and K.J.W. performed most of experiments and mainly contributed to writing the manuscript. K.J.W., S.S.K., E.B. D.V.M. J. A., S. K., D.E.C. S.N., E.B. W.X., R.S., D.V.M., M.C., and S.H.C. performed additional experiments. S.L.W., E.B., and S.N. contributed to revising the manuscript.

## Competing interests

The authors declare no competing interests.
