## [Peer Review File · Nature Communications]

Reviewers' comments:

Reviewer #1 (Remarks to the Author):

Here, the authors have performed a tour de force study demonstrating unequivocally for the first time that A β 42/40 ratio, rather than total A β or A β 42 level, dictates the creation of neurofibrillary tangle pathology in Alzheimer's disease. Using their highly innovative 3D human neural progenitor cells (hNPC) in vitro model system, they first generated clonal FAD-mutant APP and presenilin expressing hNPCs from single cells that produce stable homogeneous but different A β 42/40 ratios. Importantly, they observed that clonal hNPC 3D cultures that had high A β 42/40 ratios exhibited pathological tau acculation and phosphorylation and neurofibrillary tangles by EM. Cultures that had low A β 42/40 ratios, even though they expressed high levels of total A β , did not display pathologic tau changes. Given the possibility that overexpression of FAD mutant presenilin could cause toxic effects independent of A β , the team also made hNPC clonal lines that express APP transmembrane domain I45F or I47F artificial mutations that generate either high or low A β 42/40 ratio, respectively. Notably, only APP I45F exhibited pathologic tau phosphorylation and neurofibrillary tangle formation, confirming that high A β 42/40 ratio drives tau pathology. Finally, the authors developed an innovative 3D-plus-2D hNPC culture model to test whether the effects of high A β 42/40 ratio are cell autonomous or cell non-cell autonomous. In this model, Naïve hNPCs were plated in a thick-layer 3D culture in a central insert that was suspended above a 2D culture of APP transmembrane domain hNPCs that secreted either high or low A β 42/40 ratio. The Naïve hNPC 3D culture was in contact with conditioned media from the 2D culture but was never in direct contact with the 2D cells. Remarkably, the 3D culture exposed to high ratio I45F media showed dramatically increased phospho-tau epitope PHF1 and total tau accumulation, compared to exposure to low ratio I47F media, demonstrating non-cell autonomy. Together, the authors' results clearly demonstrate for the first time that the A β 42/40 ratio regulates tau pathology in human neuronal cells, emphasizing the importance of reducing the A β 42/40 ratio as a therapeutic strategy in AD.

The current study is highly significant and impactful for several reasons. First, to date, there has been no direct evidence for a causal role of A β 42 accumulation or elevated A β 42/40 ratio on neurofibrillary tangle formation in human neurons or mouse models. This was an enormous gap in our knowledge and definitively filling it as the authors have so done here is truly paradigm-shifting for the field. Debates on how A β 42 causes AD pathogenesis have raged ever since the discovery of the association of A β 42 with AD, and now the authors' study has convincingly settled these debates once and for all, concluding that high A β 42/40 ratio, rather than high A β 42 alone or total A β , drives pathologic tau. Another important point is the authors' demonstration of the non-cell autonomous nature of the high A β 42/40 ratio effect on causing pathogenic tau. Although much remains unanswered regarding the exact mechanism of the high A β 42/40 ratio effect, proof that it is the ratio and not absolute concentration of A β 42 that is the driving influence is a necessary first step in developing mechanistic hypotheses to test. Regardless of exact mechanism, the authors' study has already taught us that approaches to reduce the A β 42/40 ratio without affecting total A β level, such as with small molecule gamma secretase modulators, should be of prime therapeutic value for AD. Finally, the authors have undertaken enormous efforts to answer the questions posed in their study and their results are extremely robust.

Reviewer #2 (Remarks to the Author):

Summary:

This paper introduces and characterizes novel 2D and 3D cell models of Alzheimer's disease (AD) and presents evidence of the amyloid- β (A β) 42/40 ratio regulating development of tau pathology in these models. Using lentiviral transduction of human neural progenitor cells with AD-associated

mutant genes (APPSL and/or PS1 Δ E9), the authors developed multiple clonal lines with distinct A β 42/40 ratio profiles. Using these lines, the authors make the following three major claims: 1. Pathogenic tau accumulation correlates more strongly with the ratio of A β 42/40 than with either total A β or individual A β 42 or A β 40 levels across a variety of clonal lines. 2: This correlation is not driven directly by the presenilin 1 mutation as shown in clonal lines engineered to produce high or low A β 42/40 ratios without the presenilin 1 mutation. 3: This phenomenon of A β 42/40 ratio driving tau pathology can function in a non-cell-autonomous manner as shown in cocultures between transduced and naïve cells, which better mimic sporadic AD. The authors do an excellent job of discussing their data in the context of the literature and suggest that these models will be valuable investigative and drug screening tools. Importantly, their results suggest that lowering of total A β alone may not be protective against AD. Instead, their data suggests that reducing the A β 42/40 ratio using selective gamma secretase modulators may be a more effective therapeutic strategy for AD. Overall, these models and data are impressive, and the paper is quite exciting.

Overall, the claims are novel and will be of interest to others in the field. The experiments are quite convincing but further clarification and discussion is needed. Also, careful examination of the figure lettering and legends is required as there were several inconsistencies.

Major Comments:

1. The authors suggest that their non-cell-autonomous model can serve as a viable model for sporadic AD, though the development of pathology in this model is dependent on co-culture with cells expressing APP TMD mutations. Does the increase in A β 42/40 ratio reflect that observed in sporadic AD? This should be addressed in the discussion of the model's applications.
2. It seems that while APP expression is high in the E6F4 line, PS1 Δ E9 is high in the D4 and H10 cell lines. Could these differences in the expression of APP and PS1 FAD mutations drive the differences in the A β 42/40 ratios? Please discuss.
3. What are the loading controls or normalizations for the figure 4d P-tau and Total tau WB? Why is there so much more Total tau in the high A β 42/40 ratio lines?
4. Figure 5 (145F vs 1457F cell lines): What was the A β 42/40 ratio for the differentiated cells (media and 3D gels)? Was there any quantification of P-tau in these cell lines (e.g., by image analysis or WB after normalizing to TUJ1 as in the earlier studies with FAD APP mutations)?

Minor Comments:

5. Figure 1: what does the asterisk represent in 1e? Why is the WB black and white in 1d but green and black in 1e?
6. Figure 2: lettering of sub-figures needs to be corrected to be consistent with legend. There are several mix-ups. Fig 2e is missing altogether. What does the asterisk on the WB represent?
7. Looking at Figure 2 and the lentiviral vectors used (Table 1), why do the A β ratios vary so much, even within lines using the same lentiviral vector?
8. On page 5: "Analysis of APPSL-GFP and PS1 Δ E9-mCherry indicated that APPSL and PS1 Δ E9 are equally expressed in ReN-mGAP10#D4 cells as compared to the parental ReN-mGAP cells (Fig. 2a)" is unclear. Does this mean that the levels in the D4 cells were the same as the levels in the parental cells? Based on the image, it seems that it means that the GFP is equal to the mCherry in the D4 cells unlike in the parental cells. Please clarify.
9. Figure 3: g and f are reversed. Figure 3a/f: What are the unlabeled bands on the WB? Figure 3d: Please add 'in media' to the y axis label for more clarity.
10. Supplemental figure 6 legend: Why does GFP or mCherry signal indicate differentiated cells as opposed to undifferentiated cells?
11. Supplemental figure 10: The top right image appears to be at a higher magnification than the others, but the 200 nm scale bar seems to be at the same scale as the 500 nm scale bars in the other images.
12. Figure 4a: In the legend "GFP or mCherry signal indicates differentiated cells". Please explain why. Figure 4b: the 'control' bar on the graph extends over the A5 group a bit, which is a bit confusing at first glance.

13. Especially considering P-tau, how viable are these antigens after the cell has died? Would the remains of dead cells stuck in the gel show up as signal on the P-tau staining?
14. Figure 5b: The I47F line appears to have more APP and less PS1-CTF and APP-CTF than the I45F lines, and its actin loading control bands look darker and larger. Is this expected given the induced mutations? Figure 5b/c: It seems like b and c are flipped in the legend. Figure 5f: Is it possible that the antibodies are getting nonspecifically stuck in the A β plaques of the I45F line? Do plaques form in the 3D gel in this cell line?
15. Why is the ratio of A β 42/40 more important than the absolute amount of A β 42 to drive tau pathology? Any thoughts on mechanism?
16. The data show that changes in APP that produce more A β can cause tau independent of the PS1 mutation, but is it clear that the PS1 mutation does not contribute to tau pathology on its own? Is the differential tau pathology seen in the PS1 mutant lines is entirely due to the A β 42/40 ratio?
17. A thorough proofreading for typos is needed.

Reviewer #3 (Remarks to the Author):

Kwak et al. observed tau phosphorylation of neural progenitor cells (NPCs) after co-culture with APP and PS1 overexpressing cells. Although the experiments were conducted properly, their results do not present any novelty to the concept, which is already known. Furthermore, the numbers of experimental flaws and inconsistent results are confusing. Explanation of the mechanism to their findings is also missing.

Major points;

1. The authors presented tau phosphorylation of NPCs after co-culture with APP and PS1 overexpressing cells and described that the A β ratio is associated with the tau pathology. However, it is already known that the increase of A β promotes tau pathology. It is unclear how their culture system differs from experiments in which A β recombinants are added to the culture medium. Furthermore, although the authors mentioned the importance of the A β 42/40 ratio for tau pathology, it is unclear whether A β 42 is toxic or A β 40 is protective for their cultured cells. The authors should clarify these issues.
2. The authors should describe the mechanism of tau pathology promotion by A β or A β ratio.
3. The authors should evaluate the effect of γ -secretase inhibitors on their results (eg. Figure 6 data of tau phosphorylation) to strengthen their experimental results.
4. The term of "3D" is misleading. The presented culture system is a combination of 2D cultures. This should be corrected.
5. The authors utilized and compared many cell lines that present various expression levels of A β -related proteins, including the endogenous level of nicastrin or presenilin. The different status of the A β 42/40 ratio is the result of these complicated factors, which has partially originated from heterogenous transgenes in the host genome. The authors should use a site-specific genome edit, such as the CRISPR-Cas9 system, to avoid experimental and artificial heterogeneity.
6. The authors presented the relationship between the A β 42/40 ratio and subsequent tau phenotypes. However, the absolute value of the A β 42/40 ratio is from 0.1 to 0.8, which is far removed from the observation in human brain. How can the authors discuss that the presented systems and results can reflect patient pathology?

Minor points

- 1) The subtitle name after single clonal selection is confusing.

- 2) In Figure 2b, why do PS1 blots fail to detect endogenous PS1? Endogenous? A PS1 band is observed only in HReN-#AP4H1.
- 3) There is no consistency between the A β 42/40 ratio and the A β 38/40 ratio. This might be quite an artificial situation. The authors should add some discussion.
- 4) In Figure 3a, the authors should provide information about the band around 38, 62 (top panel), 28, and 50 kDa (lower panel). Why didn't the authors mention anything about the full-length APP, like in Figure 1?
- 5) In Figure 3g, the signals of the blue panel (3D6) appear localized in cytosol, which is located in the background. This panel feels misleading and should be improved to avoid being mistaken as senile plaque. In supplementary Figure 6, the blue signal is also colocalized with crowded MAP2-positive cells. We cannot rule out the possibility that the blue signal originated from nothing but the noise of scattered light, which is generally observed in neuronal cell culture.
- 6) In Figure 4, the p-Tau level is strongly affected by the dose of total tau. The authors should also use the p-Tau/total Tau ratio. The expression level of Tau is also affected by the maturation status of neural stem cells.

Reviewer #1:

Here, the authors have performed a tour de force study demonstrating unequivocally for the first time that A β 42/40 ratio, rather than total A β or A β 42 level, dictates the creation of neurofibrillary tangle pathology in Alzheimer's disease. Using their highly innovative 3D human neural progenitor cells (hNPC) in vitro model system, they first generated clonal FAD-mutant APP and presenilin expressing hNPCs from single cells that produce stable homogeneous but different A β 42/40 ratios. Importantly, they observed that clonal hNPC 3D cultures that had high A β 42/40 ratios exhibited pathological tau accumulation and phosphorylation and neurofibrillary tangles by EM. Cultures that had low A β 42/40 ratios, even though they expressed high levels of total A β , did not display pathologic tau changes. Given the possibility that overexpression of FAD mutant presenilin could cause toxic effects independent of A β , the team also made hNPC clonal lines that express APP transmembrane domain. I45F or I47F artificial mutations that generate either high or low A β 42/40 ratio, respectively. Notably, only APP I45F exhibited pathologic tau phosphorylation and neurofibrillary tangle formation, confirming that high A β 42/40 ratio drives tau pathology. Finally, the authors developed an innovative 3D-plus-2D hNPC culture model to test whether the effects of high A β 42/40 ratio are cell autonomous or cell non-cell autonomous. In this model, Naïve hNPCs were plated in a thick-layer 3D culture in a central insert that was suspended above a 2D culture of APP transmembrane domain hNPCs that secreted either high or low A β 42/40 ratio. The Naïve hNPC 3D culture was in contact with conditioned media from the 2D culture but was never in direct contact with the 2D cells. Remarkably, the 3D culture exposed to high ratio I45F media showed dramatically increased phospho-tau epitope PHF1 and total tau accumulation, compared to exposure to low ratio I47F media, demonstrating non-cell autonomy. Together, the authors' results clearly demonstrate for the first time that the A β 42/40 ratio regulates tau pathology in human neuronal cells, emphasizing the importance of reducing the A β 42/40 ratio as a therapeutic strategy in AD.

The current study is highly significant and impactful for several reasons. First, to date, there has been no direct evidence for a causal role of A β 42 accumulation or elevated A β 42/40 ratio on neurofibrillary tangle formation in human neurons or mouse models. This was an enormous gap in our knowledge and definitively filling it as the authors the authors have so done here is truly paradigm-shifting for the field. Debates on how A β 42 causes AD pathogenesis have raged ever since the discovery of the association of A β 42 with AD, and now the authors' study has convincingly settled these debates once and for all, concluding that high A β 42/40 ratio, rather than high A β 42 alone or total A β , drives pathologic tau. Another important point is the authors' demonstration of the non-cell autonomous nature of the high A β 42/40 ratio effect on causing pathogenic tau. Although much remains unanswered regarding the exact mechanism of the high A β 42/40 ratio effect, proof that it is the ratio and not absolute concentration of A β 42 that is the driving influence is a necessary first step in developing mechanistic hypotheses to test. Regardless of exact mechanism, the authors' study has already taught us that approaches to reduce the A β 42/40 ratio without affecting total A β level, such as with small molecule gamma secretase modulators, should be of prime therapeutic value for AD. Finally, the authors have undertaken enormous efforts to answer the questions posed in their study and their results are extremely robust."

We really appreciate reviewer #1 for the kind comments about the manuscript and our 3D culture models of AD. We are excited to provide our 3D culture models to AD researchers.

Reviewer #2:

Summary:

This paper introduces and characterizes novel 2D and 3D cell models of Alzheimer's disease (AD) and presents evidence of the amyloid- β ($A\beta$) 42/40 ratio regulating development of tau pathology in these models. Using lentiviral transduction of human neural progenitor cells with AD-associated mutant genes (APPSL and/or PS1 Δ E9), the authors developed multiple clonal lines with distinct $A\beta$ 42/40 ratio profiles. Using these lines, the authors make the following three major claims: 1. Pathogenic tau accumulation correlates more strongly with the ratio of $A\beta$ 42/40 than with either total $A\beta$ or individual $A\beta$ 42 or $A\beta$ 40 levels across a variety of clonal lines. 2: This correlation is not driven directly by the presenilin 1 mutation as shown in clonal lines engineered to produce high or low $A\beta$ 42/40 ratios without the presenilin 1 mutation. 3: This phenomenon of $A\beta$ 42/40 ratio driving tau pathology can function in a non-cell-autonomous manner as shown in cocultures between transduced and naïve cells, which better mimic sporadic AD. The authors do an excellent job of discussing their data in the context of the literature and suggest that these models will be valuable investigative and drug screening tools. Importantly, their results suggest that lowering of total $A\beta$ alone may not be protective against AD. Instead, their data suggests that reducing the $A\beta$ 42/40 ratio using selective gamma secretase modulators may be a more effective therapeutic strategy for AD. Overall, these models and data are impressive, and the paper is quite exciting.

Overall, the claims are novel and will be of interest to others in the field. The experiments are quite convincing but further clarification and discussion is needed. Also, careful examination of the figure lettering and legends is required as there were several inconsistencies.

Major Comments:

1. *The authors suggest that their non-cell-autonomous model can serve as a viable model for sporadic AD, though the development of pathology in this model is dependent on co-culture with cells expressing APP TMD mutations. Does the increase in $A\beta$ 42/40 ratio reflect that observed in sporadic AD?*

This is an excellent point. As the reviewer mentioned, the most of late-onset sporadic AD (SAD) patients (> 95%) do not carry familial AD mutations that increase $A\beta$ 42/40 ratio. Thus, it is an important question whether the elevated $A\beta$ 42/40 ratio also trigger the pathogenic cascade in sporadic AD patients. Indeed, previous studies showed that there is a dramatic accumulation of $A\beta$ 42 species in both FAD and SAD. According to Wang *et al.*, $A\beta$ 42/40 ratio was increased by ~3 fold both in TBS soluble and formic acid fractions in brains of SAD patients as compare to aged controls¹. Recent studies also showed that there were significant decreases in the $A\beta$ 42/40 ratio in brain Cerebrospinal fluid (CSF) at the relatively early stage of SAD, which became a standard biomarker for AD diagnosis². The dramatic decreases in the CSF $A\beta$ 42/40 ratio in AD patients are possibly caused by selective aggregation/accumulation of $A\beta$ 42 species in brain parenchyma, supporting the increase in brain $A\beta$ 42/40 ratio in SAD patients². Together, these strongly support that the increase of $A\beta$ 42/40 ratio contribute to AD pathogenesis both in FAD and SAD. Of course, further studies will be needed to confirm that the shift in $A\beta$ 42/40 ratio is driving AD pathogenesis in SAD patients. We added new paragraphs regarding these points into our revised manuscript. We appreciate the reviewer for raising this important point.

2. *It seems that while APP expression is high in the E6F4 line, PS1 Δ E9 is high in the D4 and H10 cell*

lines. Could these differences in the expression of APP and PS1 FAD mutations drive the differences in the A β 42/40 ratios? Please discuss.

Yes, this is the basic principle how we could get clonal AD cells with different A β 42/40 ratio with different A β levels. As shown in table 1 (and the new paragraph added in introduction), the levels of APPSL determine amount of total A β levels in clonal AD hNPCs while PS1 Δ E9 levels affects A β 42/40 ratio by altering PS1/ γ -secretase complex structures (Table 1). In case of hNPCs with APPSL-PS1 Δ E9 construct (HReN-mGAP and ReN-mAP4#E6F4 cells), the A β 42/40 ratio is to be fixed with total A β levels since transcription of both APPSL and PS1 Δ E9 are controlled by a single CMV promoter (Table 1). We added this paragraph at the introduction section of the revised manuscript to clarify these points.

3. *What are the loading controls or normalizations for the figure 4d P-tau and Total tau WB?*

According to the reviewer's requested, the images of β -actin bands (the loading control) of the same samples were shown in revised **Fig. 4d**. Since the samples at **Fig. 4d** were from detergent (1% sarkosyl)-insoluble fraction, most of housekeeping proteins such as β -actin or α / β -tubulin are washed away by 1% sarkosyl treatment in these samples. Instead of β -actin bands in the insoluble fraction, we re-ran the original whole sarkosyl lysates, which were used to for this study without ultracentrifugation and showed by β -actin levels, which clearly showed that the same amount of proteins was loaded into the ultracentrifugation analysis of insoluble fractions.

Why is there so much more Total tau in the high A β 42/40 ratio lines?

This is another excellent point. Since we observed elevated neuronal death in cells with high A β 42/40 ratio (**supplementary Fig. 11**), the increased level of total tau proteins in these cells cannot be explained by altered neuronal populations. Indeed, previous studies showed that total tau levels were increased by several folds in brains of AD patients as well as p-tau levels, possibly due to decreased degradation rate of hyperphosphorylated tau protein and/or elevated upregulation of tau translation³⁻⁵. In iPSC-derived human neurons with FAD mutations, increases in total tau levels also observed, which might be an early stage of tau pathology (tau proteostasis)⁶. Finally, we would like to re-emphasize that robust increases in total tau protein were mostly observed in a detergent-insoluble fractions (**Fig. 4d** and **6c**), which can be explained by elevated tau aggregation of p- and total tau together, not by total tau increases in general. Together, our observation that total tau levels were increased in the high A β 42/40 ratio lines, especially in detergent-insoluble fractions, is consistent with previous observation with AD brains and iPSC-derived cellular models from AD patients.

4. *Figure 5 (I45F vs I457F cell lines): What was the A β 42/40 ratio for the differentiated cells (media and 3D gels)? Was there any quantification of P-tau in these cell lines (e.g., by image analysis or WB after normalizing to TUJ1 as in the earlier studies with FAD APP mutations)?*

In the original manuscript, we showed A β 38, 40 and 42 levels for the differentiated cell in the media and 3D gels in **Fig. 5e**. However, the A β 42/40 ratios were not shown since some of the values were under the detection limit of the MSD assay used in these experiments (please see the table below, which contains the raw data for **Fig. 5e**). Although A β 42/40 ratios were not shown, we believe that the current graphs in **Fig. 5e** clearly showed dramatic increases of A β 42 against A β 40 in 3D gel from I45F cells as compared to I47F (**Fig. 5e**, right panel). On the contrary, A β 42 levels were relatively low in the media from I45F cells, suggesting that selective aggregation of A β 42 species in 3D gels (**Fig 5e**, left

panel). These data strongly support our conclusion that I45F selectively increased the accumulation of pathogenic A β 42 in 3D gels.

Upon reviewer's request, we added new figures that showed p-tau levels in I45F and I47F cells, as well as controls (**supplementary Fig 14a and b**). Unbiased quantitation of p-tau levels showed ~3-fold increases in p-tau levels in I45F cells as compared to I47F and also APP WT without FAD mutations. These results clearly demonstrate that high A β 42/40 ratio is correlated with p-tau pathology in 3D APP TMD cell autonomous culture system, supporting our original conclusion.

Media	WT	I47F	I45F
A β 38	8.53	37.89	118.36
	0.00	38.83	98.57
	8.53	22.00	81.99
A β 40	59.25	320.93	96.47
	49.79	290.05	77.23
	52.57	233.35	66.02
A β 42	4.51	8.49	20.54
	3.80	7.96	21.40
	3.55	5.98	15.72

3D gel	WT	I47F	I45F
A β 38	0.00	0.00	0.00
	0.00	0.00	0.00
	0.00	0.00	0.00
A β 40	0.00	57.34	0.00
	33.66	5.98	86.56
	0.00	0.00	5.98
A β 42	0.00	0.00	123.01
	0.00	0.00	140.71
	2.12	0.00	146.67

Minor Comments:

5. *Figure 1: what does the asterisk represent in 1e? Why is the WB black and white in 1d but green and black in 1e?*

The asterisk represents a non-specific band recognized by 6E10 antibody. Due to high level of bovine serum albumin in the B-27-stem cell expansion media, we generally detect non-specific bands around ~50 kDa in the media depending on the Western blot analysis condition. This band is not from human APP overexpression since the same non-specific band was detected in control GFP or mCherry cells without APP overexpression. We clarify this in the revised figure legend.

The reason for the different colors between **Fig. 1d** and **e** is that we used different detection methods for antibody-conjugated bands. For **Fig. 1d**, we use the conventional HRP-ECL method while for **Fig. 1e**, we used fluorescently labeled secondary antibody (IRDye 800CW Goat anti-Mouse IgG (P/N 926-32210), Li-COR).

6. *Figure 2: lettering of sub-figures needs to be corrected to be consistent with legend. There are several mix-ups. Fig 2e is missing altogether. What does the asterisk on the WB represent?*

These errors have been fixed in the revised figures and manuscript. We also added the meaning of asterisk in the revised figure legend. We really appreciate the reviewer for detecting these errors.

7. *Looking at Figure 2 and the lentiviral vectors used (Table 1), why do the A β ratios vary so much, even within lines using the same lentiviral vector?*

Again, this is an excellent point. As we addressed in the answer for the major point #2, the level of APPSL determines amount of total A β levels while PS1 Δ E9 levels affects A β 42/40 ratio by altering PS1/ γ -secretase complex structures (Table 1). In case of hNPCs with APPSL-PS1 Δ E9 construct, the A β 42/40 ratio could not be changed since the transcription of both APPSL and PS1 Δ E9 is controlled by a single CMV promoter (Table 1). Indeed, most clonal AD cell lines follow these rules if we compare Western blot analysis (APPSL, PS1 Δ E9 levels in **Fig. 2b**) and A β levels and ratio in **Fig. 2c and d**.

However, we found that there is some exception in this rule with HReN-#C2H2 and ReN-mAP4#E6F4 cells. The HReN-#C2H2 and ReN-mAP4#E6F4 cells express the APPSL-PS1ΔE9 construct like HReN-#A4H1 cells but we found that these cells showed much lower Aβ42/40 as compared to parental or HReN-#A4H1 cells (**Fig. 2d**). With Western blot analysis, we observed that ReN-mAP4#E6F4 cells (as well as HReN-#C2H2) expressed very low levels of PS1ΔE9 protein as compared to HReN-#A4H1 cells through an unknown compensatory mechanism, which can explain the reduced Aβ42/40 ratio in these cell lines (**Fig. 2b**). These points were added to our revised manuscript.

8. On page 5: “Analysis of APPSL-GFP and PSIDE9-mCherry indicated that APPSL and PSIDE9 are equally expressed in ReN-mGAP10#D4 cells as compared to the parental ReN-mGAP cells (Fig. 2a)” is unclear. Does this mean that the levels in the D4 cells were the same as the levels in the parental cells? Based on the image, it seems that it means that the GFP is equal to the mCherry in the D4 cells unlike in the parental cells. Please clarify.

In the ReN-mGAP cells, APPSL expression is tied with the expression of GFP since they are under the same transcriptional regulation through an IRES element (table 1). The same for mCherry and PS1ΔE9. Therefore, we interpret homogeneous overlap of GFP and mCherry in clonal D4 cells (**Fig. 2a**) as an indicator for homogeneous (not equal) expression of APPSL and PS1ΔE9 proteins in a cellular level. We agree that the sentence in our original manuscript can be somewhat mislead, and therefore, we edited these sentences in the revised manuscript. We appreciate the reviewer for raising this issue.

9. Figure 3: g and f are reversed. Figure 3a/f: What are the unlabeled bands on the WB? Figure 3d: Please add ‘in media’ to the y axis label for more clarity.

According to the reviewer’s comment, we corrected figure legends and the main text. The unidentified bands are either partially cleaved APP fragments, large Aβ, or nonspecific bands detected by 6E10 antibody, which would need further studies. We added asterisks for these bands and briefly explain these in the figure legend. In addition, we added y axis label as suggested.

10. Supplemental figure 6 legend: Why does GFP or mCherry signal indicate differentiated cells as opposed to undifferentiated cells?

Again, we agree that this sentence can be somewhat mislead. We removed this sentence in the figure legend. We appreciate the reviewer for the correction.

11. Supplemental figure 10: The top right image appears to be at a higher magnification than the others, but the 200 nm scale bar seems to be at the same scale as the 500 nm scale bars in the other images.

We replace the top right image with different one with high quality. Indeed, the scale bar was correct, but we agree with the reviewer that the lower quality of the previous image may mislead the readers about the nature of filamentous tau structures in our 3D model. We appreciate the reviewer for finding this.

12. Figure 4a: In the legend “GFP or mCherry signal indicates differentiated cells”. Please explain why. Figure 4b: the ‘control’ bar on the graph extends over the A5 group a bit, which is a bit confusing at first glance.

Again, we removed the sentence from the figure legend. We also corrected “control” bar at **Fig. 4b** accordingly.

13. Especially considering P-tau, how viable are these antigens after the cell has died? Would the remains of dead cells stuck in the gel show up as signal on the P-tau staining?

Yes, it is possible that dying and dead cells still show up p-tau signal (like a ghost cells). Indeed, we found that some of cells with strong p-tau signal do not show GFP or mCherry expression, suggesting that these cells undergo cellular deficits that cannot constitutively express GFP or mCherry proteins. It is a very interesting topic whether p-tau accumulation (and aggregation) plays a detrimental or protective role. However, this will be beyond the scope of our current manuscript and be explored in our follow up studies. We really appreciate the reviewer for raising this interesting mechanistic question.

14. Figure 5b: The I47F line appears to have more APP and less PS1-CTF and APP-CTF than the I45F lines, and its actin loading control bands look darker and larger. Is this expected given the induced mutations?

No. Bolduc *et al.* showed that I45F and I47F APP TMD mutation also differentially affect APP epsilon cleavages, which may lead to differential APP-CTF levels⁷. However, APP-CTF levels were modulated by various cellular conditions and therefore it is hard to conclude if altered PS/ γ -secretase cleavages affect APP-CTF level changes. Regarding the PS1-CTF level, it is not clear how APP TMD mutation can affect levels or maturation of γ -secretase complex, which would need additional studies. At this moment, we are afraid that this subject is beyond the scope of this manuscript. We appreciate reviewer for raising this interesting point.

Figure 5b/c: It seems like b and c are flipped in the legend.

We fixed the error. Thank you very much!

Figure 5f: Is it possible that the antibodies are getting nonspecifically stuck in the A β plaques of the I45F line? Do plaques form in the 3D gel in this cell line?

We have an extensive experience with A β and p-tau immunostaining in our 3D AD cellular models. We did not observe any cross-reactivity between A β plaque and p-tau staining in many different 3D AD cellular models. To avoid any concerns on the potential overlaps between A β and p-tau staining, we provided new high-resolution images for PHF1 staining of APP TMD neurons in supplementary Fig.14. These high-resolution images clearly showed that elevated PHF1-positive p-tau proteins were accumulated in cell bodies and neurite-like structures, not in extracellular space like A β plaques.

Do plaques form in the 3D gel in this cell line?”

Yes. We provided new immunofluorescent images of 3D6 staining to detect A β in 3D culture of APP TMD cells (**Supplementary Fig 13**). As expected, we observed robust increases in A β staining with APP I45F cells, as compared to APP I47F cells.

15. Why is the ratio of A β 42/40 more important than the absolute amount of A β 42 to drive tau

pathology? Any thoughts on mechanism?

As we discussed in the original manuscript, high A β 42/40 ratio would lead to produce more toxic A β oligomeric structures, which drives robust tau pathology. Previous studies showed that A β 40 and A β 42 form distinctive oligomeric structures⁸. Interestingly, A β 42/A β 40 mixture rapidly formed small spherical oligomers, which were more toxic than oligomeric preparations composed of either A β 40 or A β 42 alone⁹⁻¹¹. Studies also showed that A β dimers and protofibrils induce synaptotoxicity^{12, 13}. Our results with 3D non-cell-autonomous models (**Fig. 6**) also support the notion that soluble species, generated in the condition of a high A β 42/40 ratio, regulates robust tau pathology. Thus, high A β 42/40 ratio would increase soluble pathogenic A β 40-42 hybrid oligomers which bind to neurons and/or astrocytes to induce tau pathology in the 3D-differentiated neurons. The high A β 42/40 ratio can also elevate levels of insoluble A β fibrils and aggregates in 3D gels. This may directly trigger tau pathology in neurons. Further studies will be needed to clarify the roles of A β oligomers, protofibrils, other fibrillar forms and possibly other A β isoforms on inducing tau pathology in the 3D human neural cell culture models^{13, 14}.

16. The data show that changes in APP that produce more A β can cause tau independent of the PS1 mutation, but is it clear that the PS1 mutation does not contribute to tau pathology on its own? Is the differential tau pathology seen in the PS1 mutant lines is entirely due to the A β 42/40 ratio?

This is an excellent suggestion. As explained earlier, we generated a new hNPC line overexpressing PS1 Δ E9 alone (ReN-m-PS1 Δ E9) (**Supplementary Fig. 12**). All of our 3D AD cellular models express high levels of Amyloid- β precursor protein (APP) to achieve high A β levels, which is sufficient enough to trigger robust A β -driven pathogenesis. Therefore, the new ReN-m-PS1 Δ E9 model can be helpful to dissect the functional role of overexpressed PS1 Δ E9 on AD pathology in our 3D AD cellular model. Previous studies suggest that presenilin FAD mutations including Δ E9 or presenilin overexpression itself can induce multiple cellular deficits such as ER stress, altered intracellular calcium levels, and membrane protein trafficking. These cellular events might also contribute to tau pathology in our 3D AD cellular models. As shown in new **supplementary Fig. 12**, we clearly demonstrate that overexpression of PS1 Δ E9 alone does not alter the p- or total tau level as compared to control cells after 6-week of 3D-differentiation. Since we have shown robust accumulation of pathology in cells expressing both APP-Swe/Lon (APP Δ SL) and PS1 Δ E9, this result strongly supports that accumulation of pathogenic A β species is a primary player for tau pathology in the 3D human neural cell culture models of AD. Together, these provides a new and important mechanical insight regarding the role of PS1 FAD mutation on tau pathology in human neurons. We are thankful to the reviewer for this suggestion.

17. A thorough proofreading for typos is needed.

According to the reviewer's suggestion, we have thoroughly inspected the whole manuscript to reduce typos.

Reviewer #3 (Remarks to the Author):

Kwak et al. observed tau phosphorylation of neural progenitor cells (NPCs) after co-culture with APP

and PS1 overexpressing cells. Although the experiments were conducted properly, their results do not present any novelty to the concept, which is already known. Furthermore, the numbers of experimental flaws and inconsistent results are confusing. Explanation of the mechanism to their findings is also missing.

We agree that our results are consistent with one of the major hypotheses of AD that the accumulation of pathogenic A β 42 (and the elevated A β 42/40 ratio) plays a primary role in triggering AD pathology. However, we would like to re-emphasize that a direct causative relationship between the excess accumulation of pathogenic A β 42 (and A β 42/40 ratio increase) and robust tau pathology (including filamentous tau aggregation) has never been proven in human neurons before. Human iPSC-derived neurons that contain familial AD mutations failed to show A β 42-driven pathological changes. Instead, these studies suggest that the accumulation of APP C-terminal fragments (APP-CTFs), not pathogenic A β 42, is responsible for endosomal trafficking deficits and partially for the altered tau metabolism (tau proteostasis)^{6, 15}. In our opinion, these neuronal models do not achieve enough pathogenic A β accumulation, which is comparable to AD brains, and therefore, may not comprehensively recapitulate AD pathological cascades that are triggered by robust accumulation of pathogenic A β species.

In the revised manuscript, we also added new results that provide new mechanistic insights how accumulation of pathogenic A β 42 (and A β 42/40 ratio increase) induce tau pathology in 3D human cell culture model of AD. To dissect a potential contribution of FAD mutations in presenilin 1 (PS1) on AD pathology in our 3D AD cellular models (as Reviewer #2 suggested), we generated a new human neural stem cell line that overexpresses presenilin with Δ E9 mutation (ReN-m-PS1 Δ E9) alone but no overexpression of mutant APP. Previous studies suggest that presenilin FAD mutations including Δ E9 (and presenilin overexpression itself) induced multiple cellular deficits such as ER stress, altered intracellular calcium levels, and membrane protein trafficking, which might also contribute to tau pathology in our 3D AD cellular models. As we showed in new **supplementary Fig. 12**, we clearly demonstrated that overexpression of presenilin 1 with Δ E9 mutation alone does not alter the phospho or total tau levels as compared to control cells after 6-week of 3D-differentiation. Since we have shown robust accumulation of pathology in cells expressing both APP-Swe/Lon (APPSL) and PS1 Δ E9, this result strongly supports that accumulation of pathogenic A β species is a primary player for tau pathology in the 3D human neural cell culture models of AD. In the new **supplementary Fig. 5d**, we also showed the high expression of *MAPT* transcripts containing exon 10, which is necessary for producing adult 4-repeat tau isoform, in the 6-week-differentiated clonal 3D AD cellular model, which has been technical challenge of utilizing human iPSC-derived neuronal cell culture models for studying A β -derived tau pathology. Together, we strongly believe that our results will provide a new and important mechanical insight about molecular mechanisms how tau pathology is triggered in human AD neurons. We also comprehensively addressed experimental flaws (?) and inconsistent results in our answers to “point-by-point” responses below. According the reviewer’s suggestion, we added additional discussion paragraphs regarding the “explanation of mechanism” below.

Major points:

1. *The authors presented tau phosphorylation of NPCs after co-culture with APP and PS1 overexpressing cells and described that the A β ratio is associated with the tau pathology. However, it is already known that the increase of A β promotes tau pathology.*

We comprehensively addressed this point as a part of responses to major comments above.

It is unclear how their culture system differs from experiments in which A β recombinants are added to the culture medium.

Synthetic/recombinant A β peptides do not readily make pathogenic A β oligomers/aggregates. Moreover, it is extremely challenging to validate their physiological relevance in AD pathogenesis in human. Early AD studies present that treatment of recombinant/synthetic A β peptide at high concentration induces acute neuronal death, which was proposed as the first “Alzheimer’s-in-a-dish” model. However, as we explained in our manuscript, these models failed to comprehensively recapitulate pathogenic cascade of AD including gradual accumulation of pathogenic A β oligomers and aggregations and robust tau pathology driven by A β . In addition, the acute neuronal death in these models may not replicate chronic neurodegeneration observed in AD patients. That’s why transgenic AD mouse models become a standard AD model for basic and clinical studies, which can mimic chronic accumulation of pathogenic A β species in the brain, which were generated from neurons harboring human familial AD mutations. In this study, together with our previous publication¹⁶, we showed that 3D human cell culture models with familial AD mutations can recapitulate not only gradual accumulation of pathogenic A β 42 but also A β 42-driven tau pathology including robust filamentous tau accumulation, which has not been feasible in standard transgenic AD mouse models or human iPSCs models. Based on these, we would like to respectively disagree with the reviewer that addition of synthetic/recombinant A β into human neurons would be equivalent to our 3D cellular culture models of AD.

Furthermore, although the authors mentioned the importance of the A β 42/40 ratio for tau pathology, it is unclear whether A β 42 is toxic or A β 40 is protective for their cultured cells. The authors should clarify these issues.

We agree with the reviewer that it will be one of the interesting follow-up studies to dissect protective roles of A β 40. A β 40 may directly interfere the formation of subpopulation of A β 42 oligomers that trigger tau pathology or may independently regulate neuronal physiology to reduce A β 42-induced pathogenic cascade. However, these will be beyond the scope of current manuscript. As we mentioned earlier, this is the first study to prove a direct causative relationship between the elevated A β 42/40 ratio and robust tau pathology (including filamentous tau aggregation), which has never been proven in human neurons before. Previous studies with AD cellular models failed to recapitulate A β 42-driven tau pathology^{6, 15}. Thus, we strongly believe that our results will provide a new and important mechanical insight about molecular mechanism how tau pathology triggered in human AD neurons. We hope to tackle this interesting question, “toxic vs protective” in our next publication. We appreciate the reviewer for this very helpful suggestion.

2. The authors should describe the mechanism of tau pathology promotion by A β or A β ratio.

As we replied in the minor question #14 of the reviewer #2, we added a new paragraph discussing potential mechanisms how A β 42/40 ratio regulates tau pathology in our 3D culture models. Previous studies showed that A β 40 and A β 42 form distinctive oligomeric structures⁸. Interestingly, A β 42/A β 40 mixture rapidly formed small spherical oligomers, which were more toxic than oligomeric preparations composed of either A β 40 or A β 42 alone⁹⁻¹¹. Studies also showed that A β dimers and protofibrils induce synaptotoxicity^{12, 13}. Our results with 3D non-cell-autonomous models (**Fig. 6**) also support the notion that soluble species, generated in the condition of a high A β 42/40 ratio, regulates robust tau pathology.

Thus, high A β 42/40 ratio would increase soluble pathogenic A β 40-42 hybrid oligomers, bind to neurons and/or astrocyte and therefore drives tau pathology in our 3D-differentiated neurons. The high A β 42/40 ratio can also elevate levels of insoluble A β fibrils and aggregates in 3D gels, which may directly trigger tau pathology in neurons. Further studies will be needed to clarify the roles of A β oligomers, protofibrils, other fibrillar forms and possibly other A β isoforms including A β 43, in inducing tau pathology in the 3D human neural cell culture models^{13, 14}.

3. The authors should evaluate the effect of γ -secretase inhibitor on their results (eg. Figure 6 data of tau phosphorylation) to strengthen their experimental results.

According to reviewer suggestion, we performed new experiments to test the impact of A β 42/40 ratios on tau pathology using a pharmacological method. Since chronic treatment of γ -secretase inhibitors would alter neuronal differentiation by blocking notch signaling, instead, we used a γ -secretase modulator, BPN-15606, which specifically reduces A β 42/40 ratios without affecting notch signaling¹⁷. In the new **Fig. 4f** and **g**, we showed that treatment of BPN-15606 dramatically reduced pathogenic p-tau accumulation in 3D-differentiated clonal AD cells with high A β 42/40 ratio. These new results further strengthened our previous results. We appreciate the reviewer for this suggestion.

4. The term of “3D” is misleading. The presented culture system is a combination of 2D cultures. This should be corrected.

In our single-clonal 3D cell-autonomous system, all the cells were grown under 3D conditions. Of course, the gravitational force gradually moves the cells and neurite to the bottom of the 2D culture plates. However, we still see clear 3D neural networks even after 5-12 weeks of differentiation. In case of our non-cell autonomous system, we used 3D-2D hybrid culture system since we would like to make pathogenic A β species accumulate only in naïve hNPCs in 3D gels, not the cells harboring APP TMD mutations in 2D. Thus, presence of the 3D gels in naïve hNPC side was essential for developing tau pathology in this system. Therefore, we believe that “3D non-cell-autonomous models of AD” properly represents the principle of this model system.

5. The authors utilized and compared many cell lines that present various expression levels of A β -related proteins, including the endogenous level of nicastrin or presenilin. The different status of the A β 42/40 ratio is the result of these complicated factors, which has partially originated from heterogenous transgenes in the host genome. The authors should use a site-specific genome edit, such as the CRISPR-Cas9 system, to avoid experimental and artificial heterogeneity.

In the new **Fig. 4f** and **g**, we showed that treatment of BPN-15606, a γ -secretase modulator specifically reduces A β 42/40 ratios¹⁷, dramatically reduced pathogenic p-tau accumulation in 3D-differentiated clonal AD cells with high A β 42/40 ratio (according to the reviewer’s suggestion). These new results strongly support our original conclusion that tau pathology is regulated by A β 42/40 ratio, not by heterogeneous transgenes in the host genome.

Multiple studies have already shown altered A β levels and tau “proteostasis” in human iPSC-derived neurons harboring familial AD mutations^{6, 15}. Indeed, they used CRISPR-Cas9 technology to either generate isogenic control neurons from familial AD patient-derived cells or directly induce heterozygous/homozygous familial AD mutation(s) in wild-type iPSCs with isogenic genetic backgrounds, as the reviewer suggested. However, as we discussed already, these neurons failed to show

A β 42-driven pathological changes. Instead, these studies suggest that the accumulation of APP CTFs, not pathogenic A β 42, is responsible for endosomal trafficking deficits and partially for the altered tau metabolism (tau proteostasis)^{6, 15}. In our opinion, these neuronal models do not achieve enough pathogenic A β accumulation, which is comparable to AD brains, and therefore, may not comprehensively recapitulate AD pathological cascades that were triggered by robust accumulation of pathogenic A β species. In addition, 2D-differentiated iPSC-derived neurons do not express substantial amount of 4-repeat adult tau isoform, which is essential for recapitulating robust tau pathology in cellular models of AD. Therefore, we strongly believe that our models can provide a novel human cellular AD model that can be used to study the impact of pathogenic A β species and drug discovery. In addition to our clonal 3D cell autonomous models, the 3D non-cell-autonomous model can also provide more ideal models of AD, which provide robust A β 42 accumulation and A β 42-driven tau pathology in naïve human neural cells without overexpressing APP or PS1.

6. The authors presented the relationship between the A β 42/40 ratio and subsequent tau phenotypes. However, the absolute value of the A β 42/40 ratio is from 0.1 to 0.8, which is far removed from the observation in human brain. How can the authors discuss that the presented systems and results can reflect patient pathology?

It is not easy to directly compare A β 42/40 ratios between 2D undifferentiated cell culture media (“0.1 to 0.8”) and AD brains since secreted A β species undergo complicated metabolic processes in human brains including microglial clearance, degradation, aggregation, and active and passive transport through blood-brain-barriers in the brain. As shown in Fig. 3e in our original manuscript, A β 42/40 ratios in 3D gels of our AD models are in the range of 2-12. Based on a recent study¹⁸, A β 42/40 ratios in the brain of AD patients harboring various familial AD mutations are in a range of 3-9, which is quite comparable to the ratio achieved in our 3D AD models. As we have already replied in the question #1 from Reviewer #2, brain A β 42/40 ratios also increased by ~3-4 fold in brains of SAD patients. These numbers clearly support that elevated A β 42/40 ratio in our 3D models are comparable to brains of AD patients.

Figure 2. The relationship between the A β 42/A β 40 ratio and average age of onset for mutations with more than two confirmed carriers with known age of onset. Tang et al., 2018

Minor point:

1) The subline name after single clonal selection is confusing.

We really appreciate your point for these confusions. We corrected these in the revised manuscript.

2) In Figure 2b, why do PS1 blots fail to detect endogenous PS1? Endogenous? A PS1 band is observed

only in HReN-#AP4H1.

In **Fig. 2b**, we have shown the presence of endogenous PS1 C-terminal fragments (PS1-CTFs), which is a part of functional PS/ γ -secretase complex in all the cells we showed. It is well-known that it not easy to detect full-length (~45kDa) since majority of endogenous PS1 Full length proteins are immediately cleaved into functional C-terminal fragment (PS1-CTF, antibody we used recognizes this) and N-terminal fragment (PS1-NTF) to become a part of a PS/ γ -secretase complex. The band that the reviewer mentioned is not endogenous PS full-length band but the overexpressed PS1 with $\Delta E9$ splicing mutation. PS1 $\Delta E9$ lacks exon 9 region of the full length PS1 protein including the cleavage site and, therefore, mature PS1 $\Delta E9$ showed slightly decreased size as compared to full length (<https://www.alzforum.org/mutations/psen-1>). You can also see decreases in PS1-CTF and accumulation of endogenous full length PS1 in the cells highly expressing PS1 $\Delta E9$ since this mutation will replace endogenous PS1/ γ -secretase complexes. In addition, as we explained in the reviewer #2's question, we found that PS $\Delta E9$ levels were dramatically reduced in ReN-mAP4#E6F4 cells due to unknown compensatory mechanism, showing relatively lower A β 42/40 ratio. Therefore, our results are completely consistent with presenilin biology.

3) There is no consistency between the A β 42/40 ratio and the A β 38/40 ratio. This might be quite an artificial situation. The authors should add some discussion.

Previous presenilin biology studies have concluded that almost all PS1 familial AD mutations (>200) increases A β 42/40 ratio and not consistently changing A β 38/40 ratio, which is completely match with our observations in this study (<https://www.alzforum.org/mutations/psen-1>). The generation of A β 42, A β 38 and A β 40 are differential regulated through independent sequential cleavage cascades (see the panel "A" of the attached figure from Chavez-Gutierrez et al., 2012). Therefore, there is no mechanistic background how A β 42/40 and A β 38/40 ratios are connected each other. According to Chavez-Gutierrez et al., PS1 familial AD mutations also decrease 3rd and 4th cleavages of A β 48-45-42-38 (panel A), resulting in significantly decrease of A β 38/42 ratio¹⁹. Indeed, as we showed in **supplementary Fig. 3a**, the A β 38/42 ratio were significantly decreased in AD cells with PS1 $\Delta E9$ mutation, inversely correlated with PS1 $\Delta E9$ expression levels. Again, our results completely match with previous presenilin studies with non-neuronal cells.

Chavez-Gutierrez et al. *et al.*, 2012

4) In Figure 3a, the authors should provide information about the band around 38, 62 (top panel), 28, and 50 kDa (lower panel). Why didn't the authors mention anything about the full-length APP, like in Figure 1?

As we explained earlier, these bands were either proteolytic cleavage products of APP full-length protein in cells or SDS-resistant high-molecular weight A β oligomers, which is technically challenging to verify at this moment. We added this information in figure legend.

In **Fig. 1**, we used antibody recognizing full-length APP and APP-CTFs. However, in **Fig. 3a**, we used the 6E10 A β used for Western blot analysis, which recognizes A β s, APP-CTF β , sAPP α and (although it is low efficiency) full-length APP. Based on the size of ~98 kDa APP band, we thought it is highly likely the sAPP α , not full-length APP. However, considering the presence of different endogenous APP isoforms in human neuronal cells, we agree with the reviewer that this band can be a mixture of full-length APP and sAPP α . Therefore we modified figure legend accordingly. We appreciate the reviewer to point out this.

5) In Figure 3g, the signals of the blue panel (3D6) appear localized in cytosol, which is located in the background. This panel feels misleading and should be improved to avoid being mistaken as senile plaque. In supplementary Figure 6, the blue signal is also colocalized with crowded MAP2-positive cells. We cannot rule out the possibility that the blue signal originated from nothing but the noise of scattered light, which is generally observed in neuronal cell culture.

Previous studies showed that A β accumulates inside cells, which were described as “intracellular A β .” Studies also showed that intracellular A β species accumulates earlier in the neurons and may play a causative role in inducing neurodegeneration²⁰. Indeed, in our new A β staining in APP TMD cells, we also showed some overlaps with GFP-cell marker and 3D6-stained A β aggregates. Therefore, we do not think the cellular A β staining in our 3D system is not just background but may be related to intracellular A β accumulation. However, there have been technical issues to specifically detecting A β species inside the cells since most of A β antibody, theoretically detect full length APP and APP-CTF in cells, although the efficiency is low. Further studies will be needed to address this interesting hypothesis. We also observed that A β aggregates in 3D gels generally exhibit high background signals by autofluorescence, possibly due to the presence of high density β -pleated sheet structures of aggregated A β peptides. This is the case in some of the overlaps between A β staining and GFP (**supplementary Fig. 13b**, enlarged picture).

6) In Figure 4, the p-Tau level is strongly affected by the dose of total tau. The authors should also use the p-Tau/total Tau ratio. The expression level of Tau is also affected by the maturation status of neural stem cells.

In the **Supplementary Fig. 8**, we have also showed p-tau/tau ratio for clonal AD cells. As we replied in the question #4 of reviewer #2, accumulation of total tau can be a part of tau pathology in AD brains and in some iPSC-derived FAD neurons. However, we agree with the reviewer that changes of neuronal differentiation or neuron death (in our high A β 42/40 cells) can affect the conclusion of our results, since p-tau and tau proteins were specifically expressed in neuronal cells. That's why we also provided a graph showing p-tau levels that were normalized by levels of Tuj1, a total neuronal marker (**Fig. 4c**), which is we believe better indicator for tau pathology in our 3D AD models. Furthermore, in

Fig. 4d, we have shown robust increases in detergent (1% sarkosyl)-resistant or aggregated tau protein in cells exhibiting high A β 42/40 ratio, strongly supporting the results from p-tau ELISA datasets in **Fig. 4b and 4c**.

References

1. Wang, J., Dickson, D.W., Trojanowski, J.Q. & Lee, V.M. The levels of soluble versus insoluble brain Abeta distinguish Alzheimer's disease from normal and pathologic aging. *Exp Neurol* **158**, 328-337 (1999).
2. Hansson, O., Lehmann, S., Otto, M., Zetterberg, H. & Lewczuk, P. Advantages and disadvantages of the use of the CSF Amyloid beta (Abeta) 42/40 ratio in the diagnosis of Alzheimer's Disease. *Alzheimers Res Ther* **11**, 34 (2019).
3. An, W.L. *et al.* Up-regulation of phosphorylated/activated p70 S6 kinase and its relationship to neurofibrillary pathology in Alzheimer's disease. *Am J Pathol* **163**, 591-607 (2003).
4. Khatoon, S., Grundke-Iqbal, I. & Iqbal, K. Brain levels of microtubule-associated protein tau are elevated in Alzheimer's disease: a radioimmuno-slot-blot assay for nanograms of the protein. *J Neurochem* **59**, 750-753 (1992).
5. Poppek, D. *et al.* Phosphorylation inhibits turnover of the tau protein by the proteasome: influence of RCAN1 and oxidative stress. *Biochem J* **400**, 511-520 (2006).
6. Moore, S. *et al.* APP metabolism regulates tau proteostasis in human cerebral cortex neurons. *Cell reports* **11**, 689-696 (2015).
7. Bolduc, D.M., Montagna, D.R., Seghers, M.C., Wolfe, M.S. & Selkoe, D.J. The amyloid-beta forming tripeptide cleavage mechanism of gamma-secretase. *Elife* **5** (2016).
8. Sengupta, U., Nilson, A.N. & Kaye, R. The Role of Amyloid-beta Oligomers in Toxicity, Propagation, and Immunotherapy. *EBioMedicine* **6**, 42-49 (2016).
9. Bitan, G. *et al.* Amyloid beta -protein (Abeta) assembly: Abeta 40 and Abeta 42 oligomerize through distinct pathways. *Proc Natl Acad Sci U S A* **100**, 330-335 (2003).
10. Chang, Y.J. & Chen, Y.R. The coexistence of an equal amount of Alzheimer's amyloid-beta 40 and 42 forms structurally stable and toxic oligomers through a distinct pathway. *FEBS J* **281**, 2674-2687 (2014).
11. Johnson, R.D. *et al.* Single-molecule imaging reveals abeta42:abeta40 ratio-dependent oligomer growth on neuronal processes. *Biophys J* **104**, 894-903 (2013).
12. Shankar, G.M. *et al.* Amyloid-beta protein dimers isolated directly from Alzheimer's brains impair synaptic plasticity and memory. *Nat Med* **14**, 837-842 (2008).
13. O'Nuallain, B. *et al.* Amyloid beta-protein dimers rapidly form stable synaptotoxic protofibrils. *J Neurosci* **30**, 14411-14419 (2010).
14. Cline, E.N., Bicca, M.A., Viola, K.L. & Klein, W.L. The Amyloid-beta Oligomer Hypothesis: Beginning of the Third Decade. *J Alzheimers Dis* **64**, S567-S610 (2018).
15. Kwart, D. *et al.* A Large Panel of Isogenic APP and PSEN1 Mutant Human iPSC Neurons Reveals Shared Endosomal Abnormalities Mediated by APP beta-CTFs, Not Abeta. *Neuron* **104**, 256-270 e255 (2019).
16. Choi, S.H. *et al.* A three-dimensional human neural cell culture model of Alzheimer's disease. *Nature* **515**, 274-278 (2014).
17. Wagner, S.L. *et al.* Pharmacological and Toxicological Properties of the Potent Oral gamma-Secretase Modulator BPN-15606. *J Pharmacol Exp Ther* **362**, 31-44 (2017).
18. Tang, N. & Kepp, K.P. Abeta42/Abeta40 Ratios of Presenilin 1 Mutations Correlate with Clinical Onset of Alzheimer's Disease. *J Alzheimers Dis* **66**, 939-945 (2018).
19. Chavez-Gutierrez, L. *et al.* The mechanism of gamma-Secretase dysfunction in familial Alzheimer disease. *EMBO J* **31**, 2261-2274 (2012).

20. Bayer, T.A. & Wirths, O. Intracellular accumulation of amyloid-Beta - a predictor for synaptic dysfunction and neuron loss in Alzheimer's disease. *Front Aging Neurosci* **2**, 8 (2010).

REVIEWERS' COMMENTS:

Reviewer #1 (Remarks to the Author):

The authors have done an exceptional job responding to the reviewer comments and an already outstanding manuscript was improved even further. Much new data has been added to the manuscript. Nothing more needs to be performed and this manuscript should be published immediately.

Reviewer #2 (Remarks to the Author):

The authors have done an excellent job of addressing the reviewers' comments and concerns. They present a thoughtful and thorough reply to each comment, including references to support their statements. They made corrections and clarified items wherever requested. In addition, they added new data to help solidify their conclusions. These new data include the development of a new neuronal cell line expressing PS1dE9 in the absence of mutant APP and show no changes in tau. They also performed a new experiment testing the effects of a gamma secretase modulator (which should influence Abeta 42/42 ratios) on tau pathology. Importantly, the authors have addressed mechanistic questions raised by the reviewers. Overall, the paper is improved. I agree with Reviewer 1 that this paper is a Tour de Force. It encompasses a huge amount of work, novel findings, and it will have a big impact in the field.

As a reviewer, I am very grateful to the authors for their comprehensive replies to my comments and suggestions.

Reviewer #3 (Remarks to the Author):

The authors responded to this reviewer's comments appropriately.